# CUDA: CURRICULUM OF DATA AUGMENTATION FOR LONG-TAILED RECOGNITION

**Sumyeong Ahn**[∗]**, Jongwoo Ko**[∗]**, Se-Young Yun**
KAIST AI
Seoul, Korea
{sumyeongahn, jongwoo.ko, yunseyoung}@kaist.ac.kr

## ABSTRACT

Class imbalance problems frequently occur in real-world tasks, and conventional deep learning algorithms are well known for performance degradation on imbalanced training datasets. To mitigate this problem, many approaches have aimed to balance among given classes by *re-weighting* or *re-sampling* training samples. These re-balancing methods increase the impact of minority classes and reduce the influence of majority classes on the output of models. However, the extracted representations may be of poor quality owing to the limited number of minority samples. To handle this restriction, several methods have been developed that increase the representations of minority samples by leveraging the features of the majority samples. Despite extensive recent studies, no deep analysis has been conducted on determination of classes to be augmented and strength of augmentation has been conducted. In this study, we first investigate the correlation between the degree of augmentation and class-wise performance, and find that the proper degree of augmentation must be allocated for each class to mitigate class imbalance problems. Motivated by this finding, we propose a simple and efficient novel curriculum, which is designed to find the appropriate per-class strength of data augmentation, called CUDA: **CU**rriculum of **D**ata **A**ugmentation for long-tailed recognition. CUDA can simply be integrated into existing long-tailed recognition methods. We present the results of experiments showing that CUDA effectively achieves better generalization performance compared to the state-of-the-art method on various imbalanced datasets such as CIFAR-100-LT, ImageNet-LT, and iNaturalist 2018. [1]

## 1 INTRODUCTION

Deep neural networks (DNNs) have significantly improved over the past few decades on a wide range of tasks (He et al., 2017; Redmon & Farhadi, 2017; Qi et al., 2017). This effective performance is made possible by come from well-organized datasets such as MNIST (LeCun et al., 1998), CIFAR-10/100 (Krizhevsky et al., 2009), and ImageNet (Russakovsky et al., 2015). However, as Van Horn et al. (2018) indicated, gathering such balanced datasets is notoriously difficult in real-world applications. In addition, the models perform poorly when trained on an improperly organized dataset, *e.g.,* in cases with class imbalance, because minority samples can be ignored due to their small portion.

The simplest solution to the class imbalance problem is to prevent the model from ignoring minority classes. To improve generalization performance, many studies have aimed to emphasize minority classes or reduce the influence of the majority samples. Reweighting (Cao et al., 2019; Menon et al., 2021) or resampling (Buda et al., 2018; Van Hulse et al., 2007) are two representative methods that have been frequently applied to achieve this goal. (i) *Reweighting* techniques increase the weight of the training loss of the samples in the minority classes. (ii) *Resampling* techniques reconstruct a class-balanced training dataset by upsampling minority classes or downsampling majority classes.

Although these elaborate rebalancing approaches have been adopted in some applications, limited information on minority classes due to fewer samples remains problematic. To address this issue, some works have attempted to spawn minority samples by leveraging the information of the minority

---

[∗]Two authors contribute equally
[1]Code is available at Link

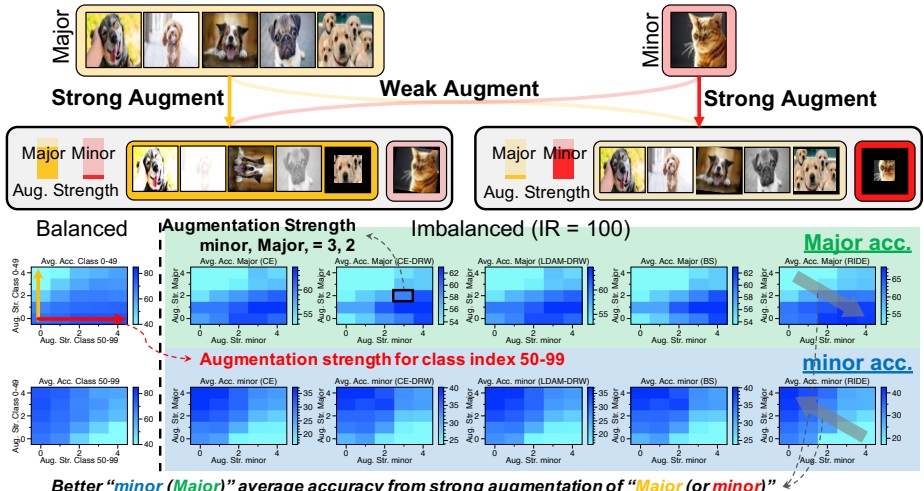

Figure 1: Motivation of CUDA. If one half of the classes (*e.g.,* class indices in 0-49) are strongly augmented, the averaged accuracy of the other half of the classes (*i.e.,* in 50-99) increases. The heatmaps in the first and the second rows present the accuracy for class indices in 0-49 and 50-99, respectively. Also, each point in the plots means the accuracy under corresponding augmentation strength. This phenomenon is also observed in the imbalanced case in which the top first half (major; the first row) have more samples than the second half classes (minor; the second row). Setup and further analysis are described in Appendix A.

samples themselves. For example, Chawla et al. (2002); Ando & Huang (2017) proposed a method to generate interpolated minority samples. Recently (Kim et al., 2020; Chu et al., 2020; Park et al., 2022) suggested enriching the information of minority classes by transferring information gathered from majority classes to the minority ones. For example, Kim et al. (2020) generated a balanced training dataset by creating adversarial examples from the majority class to consider them as minority.

Although many approaches have been proposed to utilize data augmentation methods to generate various information about minority samples, relatively few works have considered the influence of the degree of augmentation of different classes on class imbalance problems. In particular, few detailed observations have been conducted as to which classes should be augmented and how intensively.

To this end, we first consider that controlling the strength of class-wise augmentation can provide another dimension to mitigate the class imbalance problem. In this paper, we use the number of augmentation operations and their magnitude to control the extent of the augmentation, which we refer to herein as its strength, *e.g.,* a strength parameter of 2 means that two randomly sampled operations with a pre-defined magnitude index of 2 are used.

Our key finding is that class-wise augmentation improves performance in the non-augmented classes while that for the augmented classes may not be significantly improved, and in some cases, performances may even decrease. As described in Figure 1, regardless of whether a given dataset is class imbalanced, conventional class imbalance methods show similar trends: when only the major classes are strongly augmented (*e.g.,* strength 4), the performance of majority classes decreases, whereas that for the minority classes have better results. To explain this finding, we further find that strongly augmented classes get diversified feature representation, preventing the growth of the norm of a linear classifier for corresponding classes. As a result, the softmax outputs of the strongly augmented classes are reduced, and thus the accuracy of those classes decreases. It is described in Appendix A. This result motivates us to find the proper augmentation strength for each class to improve the performance for other classes while maintaining its own performance.

**Contribution.** We propose a simple algorithm called **CU**rriculum of **D**ata **A**ugmentation (CUDA) to find the proper class-wise augmentation strength for long-tailed recognition. Based on our motivation, we have to increase the augmentation strength of majorities for the performance of minorities when the model successfully predicts the majorities. On the other hand, we have to lower the strength of majorities when the model makes wrong predictions about majorities. The proposed method consists of two modules, which compute a level-of-learning score for each class and leverage the score to determine the augmentation. Therefore, CUDA increases and decreases the augmentation

strength of the class that was successfully and wrongly predicted by the trained model. To the best of our knowledge, this work is the first to suggest a class-wise augmentation method to find a proper augmentation strength for class imbalance problem.

We empirically examine performance of CUDA on synthetically imbalanced datasets such as CIFAR-100-LT (Cao et al., 2019), ImageNet-LT (Liu et al., 2019), and a real-world benchmark, iNaturalist 2018 (Van Horn et al., 2018). With the high compatibility of CUDA, we apply our framework to various long-tailed recognition methods and achieve better performance compared to the existing long-tailed recognition methods. Furthermore, we conduct an extensive exploratory analysis to obtain a better understanding of CUDA. The results of these analyses verify that CUDA exhibits two effects that mitigate class imbalance, including its balanced classifier and improved feature extractor.

## 2 RELATED WORKS

**Long-tailed Recognition (LTR).** The datasets with class imbalances can lead DNNs to learn biases toward training data, and their performance may decrease significantly on the balanced test data. To improve the robustness of such models to imbalance, LTR methods have been evolving in two main directions: (1) reweighting (Cui et al., 2019; Cao et al., 2019; Park et al., 2021) methods that reweight the loss for each class by a factor inversely proportional to the number of data points, and (2) resampling methods (Kubat et al., 1997; Chawla et al., 2002; Ando & Huang, 2017) that balance the number of training samples for each class in the training set. However, studies along these lines commonly sacrifice performance on majority classes to enhance that on minority classes, because the overfitting problem occurs with limited information on minority classes as a result of increasing the weight of a small number of minority samples.

Several methods have recently been developed to alleviate the overfitting issues in various categories: (1) two-stage training (Cao et al., 2019; Kang et al., 2020; Liu et al., 2019), (2) ensemble methods (Zhou et al., 2020a; Xiang et al., 2020; Wang et al., 2021; Cai et al., 2021), and (3) contrastive learning approach (Kang et al., 2021; Cui et al., 2021; Zhu et al., 2022; Li et al., 2022a;b). To re-balance the classifier layers after achieving a good representation on the imbalanced training dataset in an early phase, Cao et al. (2019) proposed deferred resampling (DRS) and reweighting (DRW) approaches. Kang et al. (2020) decoupled the learning procedure into representation learning and training linear classifier, achieved higher performance than previous balancing methods. Wang et al. (2021) and Cai et al. (2021) suggested efficient ensemble methods using multiple experts with a routing module and a shared architecture for experts to capture various representations. Liu et al. (2022) found that self-supervised representations are more robust to class imbalance than supervised representations, and some works have developed supervised contrastive learning methods (Khosla et al., 2020) for imbalanced datasets (Cui et al., 2021; Zhu et al., 2022; Li et al., 2022b).

Another line of research has considered augmentation methods in terms of both input and feature spaces (Kim et al., 2020; Chu et al., 2020; Li et al., 2021). Recently, Park et al. (2022) mixed minority and majority images by using CutMix with different sampling strategies to enhance balancing and robustness simultaneously. These methods commonly focus on utilizing the rich context of majority samples to improve the diversity of minority samples. Zhou et al. (2022) proposed an augmentation-based contrastive learning method which boosts memorization of each samples for long-tailed learning. Moreover, these augmentation-based methods are relatively in easy to apply orthogonally with other LTR methods.

**Data Augmentation (DA).** DA has been studied to mitigate overfitting which may occur due to a lack of data samples. Some works have been proposed to erase random parts of images to enhance the generalization performance of neural networks (DeVries & Taylor, 2017; Zhong et al., 2020; Kumar Singh & Jae Lee, 2017; Choe & Shim, 2019). Recently, variants of MixUp (Zhang et al., 2018) have been proposed; this method combines two images with specific weights (Tokozume et al., 2018; Guo et al., 2019; Takahashi et al., 2018; DeVries & Taylor, 2017; Verma et al., 2019). By aggregating two approaches, CutMix (Yun et al., 2019) was proposed to erase and replace a small rectangular part of an image into another image. In another line of research, methods have been proposed to automatically configure augmentation operations (Cubuk et al., 2019; Lim et al., 2019; Li et al., 2020b; Hataya et al., 2020; Gudovskiy et al., 2021). In addition, Cubuk et al. (2020) randomly selected augmentation operations using the given hyperparameters of the number of sampling augmentation and their magnitudes. Recently, class-wise or per-sample auto-augmentation methods have also been proposed (Cheung & Yeung, 2021; Rommel et al., 2022).

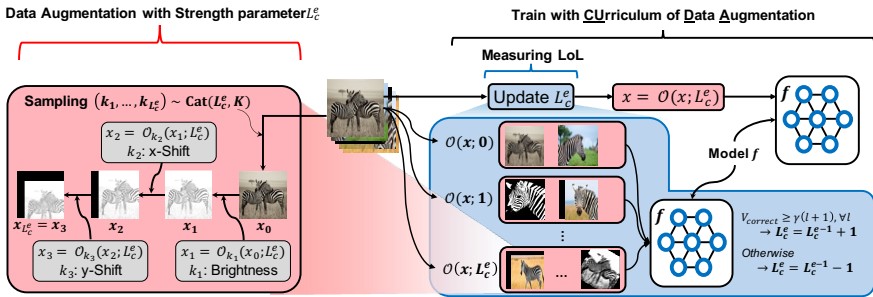

Figure 2: Algorithm overview. CUDA is composed of two main parts: (1) strength-based augmentation and (2) Level-of-Learning (LoL) score. To control the difficulty of augmented images, strength-based augmentation utilizes two values, the number of augmentations and their magnitudes. We use the strength-based augmentation module to score the LoL. Based on the measured LoL score, CUDA generates adequately augmented images for LTR algorithms.

## 3  CURRICULUM OF DATA AUGMENTATION FOR LONG-TAILED RECOGNITION

The core philosophy of CUDA is to *"generate an augmented sample that becomes the most difficult sample without losing its original information."* In this section, we describe design of CUDA in terms of two parts: (1) a method to generate the augmented samples based on the given strength parameter, and (2) a method to measure a Level-of-Learning (LoL) score for each class.

### 3.1  PROBLEM FORMULATION OF LONG-TAILED RECOGNITION

Suppose that the training dataset $\mathcal{D} = \{(x_i, y_i)\}_{i=1}^{N}$ is composed of images with size $d$, $x_i \in \mathbb{R}^d$, and their corresponding labels $y_i \in \{1, ..., C\}$. $\mathcal{D}_c \subset \mathcal{D}$ is a set of class $c$, *i.e.*, $\mathcal{D}_c = \{(x, y) | y = c, (x, y) \in \mathcal{D}\}$. Without loss of generality, we assume $|\mathcal{D}_1| \geq |\mathcal{D}_2| \geq \cdots \geq |\mathcal{D}_C|$, where $|\mathcal{D}|$ denotes the cardinality of the set $\mathcal{D}$. We denote the $N_{\max} := |\mathcal{D}_1|$ and $N_{\min} := |\mathcal{D}_C|$. LTR algorithms, $\mathcal{A}_{\text{LTR}}(f_\theta, \mathcal{D})$, mainly focus on training the model $f_\theta$ with parameter $\theta$ when the class distribution of training dataset $\mathcal{P}_{\text{train}}(y)$ and test dataset $\mathcal{P}_{\text{test}}(y)$ are not identical. More precisely, $\mathcal{P}_{\text{train}}(y)$ is highly imbalanced while $\mathcal{P}_{\text{test}}(y)$ is balanced, *i.e.,* uniform distribution.

### 3.2  CURRICULUM OF DATA AUGMENTATION

In this section, we describe our proposed DA with strength parameter, and the methods used to measured the LoL score. Then, we integrate the two methods in a single framework to propose CUDA.

**DA with a strength parameter.** Let us assume that there exist pre-defined $K$ augmentation operations. We utilize visual augmentation operations which is indexed as $k \in \{1, \cdots, K\}$, *e.g.*, Gaussian blur, Rotation, Horizontal flip. Each augmentation operation $\mathcal{O}_k^{m_k(s)} : \mathbb{R}^d \to \mathbb{R}^d$ has its own pre-defined augmentation magnitude function $m_k(s)$ where the strength parameter $s \in \{0, ..., S\}$. These operations are described in detail along with each magnitude functions in Appendix D.

Given an augmentation strength parameter $s$ and an input image $x$, we model a sequence of augmentation operations $\mathcal{O}(x; s)$ as follows:

$$\mathcal{O}(x; s) = \mathcal{O}_{k_s}^{m_{k_s}(s)} \circ \mathcal{O}_{k_{s-1}}^{m_{k_{s-1}}(s)} \circ \cdots \circ \mathcal{O}_{k_1}^{m_{k_1}(s)}(x), \quad k_i \sim \text{Cat}(K, \mathcal{U}(K)) \quad \forall i = \{1, \ldots, s\},$$

where, $\text{Cat}(\cdot)$ and $\mathcal{U}(\cdot)$ denote categorical and discrete uniform distributions, respectively. The sequential augmentation operation $\mathcal{O}(x; s)$ samples $s$ operations from the categorical distribution when the probability of seeing the operations follows uniform distribution. As depicted on the left side Figure 2, suppose that the random sampled augmentations $k_1$, $k_2$, and $k_3$ are brightness, X-shift, and Y-shift, respectively. Then, $\mathcal{O}(x; 3)$ outputs an image in which bright is raised by $m_{\text{bright}}(3)$ and moved by $m_{\text{x-shift}}(3)$ on the x-axis and shifted by $m_{\text{y-shift}}(3)$ on the y-axis.

| **Algorithm 1: CUrriculum of Data Augmentation** | **Algorithm 2:** $V_{\text{LoL}}$: Update LoL score |
|---|---|
| **Input:** LTR algorithm $\mathcal{A}_{\text{LTR}}(f, \mathcal{D})$, training dataset $\mathcal{D} = \{(x_i, y_i)\}_{i=1}^N$, train epochs $E$, aug. probability $p_{\text{aug}}$, threshold $\gamma$, number of sample coefficient $T$. 
 **Output:** trained model $f_\theta$ 
 **Initialize:** $L_c^0 = 0 \ \forall c \in \{1, ..., C\}$ 
 **for** $e \le E$ **do** 
    Update $L_c^e = V_{\text{LoL}}(\mathcal{D}_c, L_c^{e-1}, f_\theta, \gamma, T) \quad \forall c$    // Alg. 2 
    Generate $\mathcal{D}_{\text{CUDA}} = \{(\bar{x}_i, y_i) \| (x_i, y_i) \in \mathcal{D}\}$ where 

      $\bar{x}_i = \begin{cases} \mathcal{O}(x_i, L_{y_i}^e) & \text{with prob. } p_{\text{aug}} \\ x_i & \text{otherwise.} \end{cases}$ 

    Run LTR algorithm using $\mathcal{D}_{\text{CUDA}}$, *i.e.,* $\mathcal{A}_{\text{LTR}}(f_\theta, \mathcal{D}_{\text{CUDA}})$. 
 **end** | **Input:** $\mathcal{D}_c, L, f_\theta, \gamma, T$ 
 **Output:** updated $L$ 
 **Initialize:** check $= 1$ 
 **for** $l \le L$ **do** 
    /* $V_{\text{correct}}(\mathcal{D}_c, l, f_\theta, T)$ */ 
    Sample $\mathcal{D}_c' \subset \mathcal{D}_c$ s.t. $\|\mathcal{D}_c'\| = T(l+1)$ 
    Compute $v = \sum_{x \in \mathcal{D}_c'} \mathbb{1}_{\{f(\mathcal{O}(x;l))=c\}}$ 
    **if** $v \le \gamma T(l+1)$ **then** 
      \| check $\leftarrow 0$; **break** 
    **end** 
 **end** 
 **if** check $= 1$ **then** $L \leftarrow L + 1$ 
 **else** $L \leftarrow L - 1$ |

**Level-of-Learning (LoL).** To control the strength of augmentation properly, we check whether the model can correctly predict augmented versions without losing the original information. To enable this, we define the LoL for each class $c$ at epoch $e$, *i.e.,* $L_c^e$, which is adaptively updated as the training continues as follows:

$$L_c^e = V_{\text{LoL}}(\mathcal{D}_c, L_c^{e-1}, f_\theta, \gamma, T),$$

where

$$V_{\text{LoL}}(\mathcal{D}_c, L_c^{e-1}, f_\theta, \gamma, T) = \begin{cases} L_c^{e-1} + 1 & \text{if } V_{\text{Correct}}(\mathcal{D}_c, l, f_\theta, T) \ge \gamma T(l+1) \quad \forall l \in \{0, ..., L_c^{e-1}\} \\ L_c^{e-1} - 1 & \text{otherwise} \end{cases} \quad .$$

Here, $\gamma \in [0, 1]$ is threshold hyperparameter, $T$ is coefficient of the number of samples used to updating LoL. $V_{\text{correct}}$ is a function which outputs the number of correctly predicted examples by the model $f_\theta$ among $l + 1$ randomly augmented samples with strength $l$. $V_{\text{correct}}$ is defined as:

$$V_{\text{Correct}}(\mathcal{D}_c, l, f_\theta, T) = \sum_{x \in \mathcal{D}_c'} \mathbb{1}_{\{f_\theta(\mathcal{O}(x;l))=c\}} \quad \text{where } \mathcal{D}_c' \subset \mathcal{D}_c.$$

Note that $\mathcal{D}_c'$ is a randomly sampled subset of $\mathcal{D}_c$ with replacement and its size is $T(l+1)$.

The key philosophy of this criterion is two fold. (1) If samples in the class $c$ are trained sufficiently with an augmentation strength of $L_c^e$, the model is ready to learn a more difficult version with augmentation strength of $L_c^{e+1} \leftarrow L_c^e + 1$. In contrast, if the model predicts incorrectly, it should re-learn the easier sample with an augmentation strength of $L_c^{e+1} \leftarrow L_c^e - 1$. (2) As the strength parameter increases, the number of candidates for the sequential augmentation operation $\mathcal{O}(x; L)$ increases exponentially. For example, the amount of increment is $N^L(N-1)$ when $L$ is increases to $L + 1$. To control the LoL in a large sequential augmentation operation space, we take more random samples to check as the strength parameter gets bigger. In our experiments, linearly increasing the number of samples to evaluate corresponding to the strength with a small additional computation time was sufficient. $V_{\text{LoL}}$ is described in Figure 2 and Algorithm 2.

**Curriculum of DA.** By combining two components, including DA with a strength parameter and LoL, our CUDA provides class-wise adaptive augmentation to enhance the performance of the others without losing its own information. As shown in Figure 2 and Algorithm 1, we measure the LoL score $L_c$ for all classes in the training dataset to determine the augmentation strength for every epoch. Based on $L_c$, we generate the augmented version $\mathcal{O}(x; L_c)$ for $x \in \mathcal{D}_c$ and train the model with the augmented samples. Additionally, we randomly use the original sample instead of the augmented sample with probability $p_{\text{aug}}$ so that the trained models do not forget the original information. In our experiments, this operation improved performance robustly on a wide range of $p_{\text{aug}}$ values. The results are provided in Section 4.3.

**Advantage of CUDA design.** Our proposed approach mainly has three advantages. (1) CUDA adaptively finds proper augmentation strengths for each class without need for a validation set. (2) Following the spirits of existing curriculum learning methods (Hacohen & Weinshall, 2019; Zhou et al., 2020b; Wu et al., 2021), CUDA enables modeling by first presenting easier examples earlier during training to improve generalization. This encourages the model to learn difficult samples (*i.e.,* within high augmentation strength) better. (3) Moreover, owing to the universality of data augmentation, CUDA is easily compatible with other LTR algorithms, such as (Cao et al., 2019; Ren et al., 2020; Wang et al., 2021).

Table 1: Validation accuracy on CIFAR-100-LT dataset. † are from Park et al. (2022) and ‡, ⋆ are from the original papers (Kim et al., 2020; Zhu et al., 2022). Other results are from our implementation. We format the first and second best results as **bold** and underline . We report the average results of three random trials.

| Algorithm | Imbalance Ratio (IR) | | | Statistics (IR 100) | | |
|---|---|---|---|---|---|---|
| | 100 | 50 | 10 | Many | Med | Few |
| CE | $38.7_{\pm0.4}$ | $43.4_{\pm0.3}$ | $56.5_{\pm0.6}$ | $66.2_{\pm0.5}$ | $37.3_{\pm0.6}$ | $8.2_{\pm0.3}$ |
| CE + CMO (Park et al., 2022) | $42.0_{\pm0.4}$ | $47.0_{\pm0.5}$ | $60.0_{\pm0.4}$ | $69.1_{\pm0.4}$ | $41.2_{\pm0.6}$ | $11.3_{\pm0.7}$ |
| CE + CUDA | $42.7_{\pm0.4}$ | $47.2_{\pm0.4}$ | $59.6_{\pm0.4}$ | $\mathbf{71.6_{\pm0.6}}$ | $42.3_{\pm0.3}$ | $9.4_{\pm0.7}$ |
| CE + CMO + CUDA | $43.5_{\pm0.5}$ | $48.7_{\pm0.6}$ | $60.0_{\pm0.3}$ | $70.0_{\pm0.7}$ | $43.4_{\pm0.5}$ | $12.7_{\pm0.8}$ |
| CE-DRW (Cao et al., 2019) | $41.4_{\pm0.2}$ | $45.5_{\pm0.6}$ | $57.8_{\pm0.6}$ | $62.8_{\pm0.5}$ | $41.7_{\pm0.7}$ | $16.1_{\pm0.4}$ |
| CE-DRW + Remix (Chou et al., 2020)† | 45.8 | 49.5 | 59.2 | - | - | - |
| CE-DRW + CUDA | $47.7_{\pm0.4}$ | $52.4_{\pm0.5}$ | $61.6_{\pm0.5}$ | $64.3_{\pm0.4}$ | $49.2_{\pm0.6}$ | $26.7_{\pm0.6}$ |
| LDAM-DRW (Cao et al., 2019) | $42.5_{\pm0.2}$ | $47.4_{\pm0.5}$ | $57.6_{\pm0.1}$ | $62.8_{\pm0.5}$ | $42.3_{\pm0.6}$ | $19.0_{\pm0.7}$ |
| LDAM + M2m (Kim et al., 2020)‡ | 43.5 | - | 57.6 | - | - | - |
| LDAM-DRW + CUDA | $47.6_{\pm0.7}$ | $51.1_{\pm0.4}$ | $58.4_{\pm0.1}$ | $67.3_{\pm0.6}$ | $50.4_{\pm0.5}$ | $21.4_{\pm0.2}$ |
| BS (Ren et al., 2020) | $43.3_{\pm0.4}$ | $46.9_{\pm0.2}$ | $58.3_{\pm0.4}$ | $61.6_{\pm0.8}$ | $42.3_{\pm0.5}$ | $23.0_{\pm0.4}$ |
| BS + CUDA | $47.7_{\pm0.3}$ | $52.1_{\pm0.4}$ | $61.7_{\pm0.5}$ | $63.3_{\pm0.4}$ | $48.4_{\pm0.4}$ | $28.7_{\pm0.5}$ |
| RIDE (3 experts) (Wang et al., 2021)† | 48.6 | 51.4 | 59.8 | - | - | - |
| RIDE (3 experts) | $49.7_{\pm0.2}$ | $52.7_{\pm0.1}$ | $60.2_{\pm0.2}$ | $67.7_{\pm0.6}$ | $51.5_{\pm0.5}$ | $26.7_{\pm0.6}$ |
| RIDE + CMO (Park et al., 2022)† | 50.0 | 53.0 | 60.2 | - | - | - |
| RIDE + CMO | $49.9_{\pm0.1}$ | $53.0_{\pm0.1}$ | $58.9_{\pm0.3}$ | $67.3_{\pm0.3}$ | $51.3_{\pm0.6}$ | $28.1_{\pm0.4}$ |
| RIDE (3 experts) + CUDA | $50.7_{\pm0.2}$ | $53.7_{\pm0.4}$ | $60.2_{\pm0.1}$ | $69.2_{\pm0.3}$ | $52.8_{\pm0.2}$ | $27.3_{\pm0.8}$ |
| BCL (Zhu et al., 2022)⋆ | $\underline{51.0}$ | $\underline{54.9}$ | $\underline{64.4}$ | 67.2 | $\underline{53.1}$ | $\underline{32.9}$ |
| BCL + CUDA | $\mathbf{52.3_{\pm0.2}}$ | $\mathbf{56.2_{\pm0.4}}$ | $\mathbf{64.6_{\pm0.1}}$ | $66.4_{\pm0.2}$ | $\mathbf{54.2_{\pm0.6}}$ | $\mathbf{33.9_{\pm0.8}}$ |

# 4 EXPERIMENTS

In this section, we present empirical evaluation, the results of which demonstrate the superior performance of our proposed algorithm for class imbalance. We first describe the long-tailed classification benchmarks and implementations in detail (Section 4.1). Then, we describe the experimental results on several synthetic (CIFAR-100-LT, ImageNet-LT) and real-world (iNaturalist 2018) long-tailed benchmark datasets in Section 4.2. Moreover, we conduct additional experiments to obtain a better understanding of CUDA, and this analysis is provided in Section 4.3.

## 4.1 EXPERIMENTAL SETUP

**Datasets.** We evaluate CUDA on the most commonly used long-tailed image classification tasks: CIFAR-100-LT (Cao et al., 2019), ImageNet-LT (Liu et al., 2019), and iNaturalist 2018 (Van Horn et al., 2018). CIFAR-100-LT and ImageNet-LT are provided with imbalanced classes by synthetically sampling the training samples. CIFAR-100-LT is examined with various imbalance ratios $\{100, 50, 10\}$, where an imbalance ratio is defined as $N_{\max}/N_{\min}$. iNaturalist 2018 is a large-scale real-world dataset includes natural long-tailed imbalance. We utilize the officially provided datasets.

**Baselines.** We compare CUDA with previous long-tailed learning algorithms , including cross-entropy loss (CE), two-stage approaches: CE-DRW (Cao et al., 2019) and cRT (Kang et al., 2020), balanced loss approaches: LDAM-DRW (Cao et al., 2019) and Balanced Softmax (BS; Ren et al. 2020), the ensemble method: RIDE with three experts (Wang et al., 2021), resampling algorithms: Remix (Chou et al., 2020) and CMO (Park et al., 2022), and contrastive learning-based approach: BCL (Zhu et al., 2022). We integrate CUDA with CE, CE-DRW, LDAM-DRW, BS, RIDE, and BCL algorithms. For longer epochs, we compare CUDA with PaCo (Cui et al., 2021), BCL, and NCL (Li et al., 2022a), by combining CUDA with BCL and NCL. For a fair comparison of the computational cost, we train the network with the official one-stage implementation of RIDE (*i.e.,* without distillation and routing).

**Implementation.** For CIFAR-100-LT dataset, almost all implementations follow the general setting from Cao et al. (2019), whereas cRT (Kang et al., 2020), BCL, NCL and RIDE follow the settings used in their original implementation. Following Cao et al. (2019), we use ResNet-32 (He et al., 2016) as a backbone network for CIFAR-100-LT. The network is trained on SGD with a momentum of 0.9 and a weight decay of $2 \times 10^{-4}$. The initial learning rate is 0.1 and a linear learning rate warm-up is used in the first 5 epochs to reach the initial learning rate. During training over 200 epochs, the learning rate is decayed at the 160th and 180th epochs by 0.01. For the ImageNet-LT and iNaturalist, the ResNet-50 is used as a backbone network and is trained for 100 epochs. The learning rate is decayed at the 60th and 80th epochs by 0.1. As with CIFAR, for cRT, RIDE, and BCL, we follow the original experimental settings of the official released code. For the hyperparameter values of

Table 2: Validation accuracy on ImageNet-LT and iNaturalist 2018 datasets. † indicates reported results from the Park et al. (2022) and ‡ indicates those from the original paper (Kang et al., 2020). ⋆ means we train the network with the official code in an one-stage RIDE.

| Algorithm | ImageNet-LT | | | | iNaturalist 2018 | | | |
|---|---|---|---|---|---|---|---|---|
| | Many | Med | Few | All | Many | Med | Few | All |
| CE† | 64.0 | 33.8 | 5.8 | 41.6 | 73.9 | 63.5 | 55.5 | 61.0 |
| CE + CUDA | $\mathbf{67.2}_{\pm 0.1}$ | $47.0_{\pm 0.2}$ | $13.5_{\pm 0.3}$ | $47.3_{\pm 0.2}$ | $\mathbf{74.6}_{\pm 0.3}$ | $64.9_{\pm 0.1}$ | $57.2_{\pm 0.1}$ | $62.5_{\pm 0.2}$ |
| CE-DRW (Cao et al., 2019) | $61.7_{\pm 0.1}$ | $47.1_{\pm 0.3}$ | $29.0_{\pm 0.3}$ | $50.1_{\pm 0.1}$ | $68.2_{\pm 0.2}$ | $67.3_{\pm 0.2}$ | $66.4_{\pm 0.1}$ | $67.0_{\pm 0.1}$ |
| CE-DRW + CUDA | $61.8_{\pm 0.1}$ | $48.3_{\pm 0.1}$ | $30.3_{\pm 0.2}$ | $51.0_{\pm 0.1}$ | $68.8_{\pm 0.1}$ | $68.1_{\pm 0.3}$ | $66.6_{\pm 0.2}$ | $67.5_{\pm 0.1}$ |
| LWS (Kang et al., 2020)‡ | 57.1 | 45.2 | 29.3 | 47.7 | 65.0 | 66.3 | 65.5 | 65.9 |
| cRT (Kang et al., 2020)‡ | 58.8 | 44.0 | 26.1 | 47.3 | 69.0 | 66.0 | 63.2 | 65.2 |
| cRT + CUDA | $62.3_{\pm 0.1}$ | $47.2_{\pm 0.2}$ | $28.4_{\pm 0.5}$ | $50.2_{\pm 0.2}$ | $68.2_{\pm 0.1}$ | $67.8_{\pm 0.2}$ | $66.4_{\pm 0.1}$ | $67.3_{\pm 0.1}$ |
| LDAM-DRW (Cao et al., 2019)† | 60.4 | 46.9 | 30.7 | 49.8 | - | - | - | 66.1 |
| LDAM-DRW + CUDA | $63.1_{\pm 0.1}$ | $48.0_{\pm 0.3}$ | $31.1_{\pm 0.2}$ | $51.4_{\pm 0.1}$ | $67.8_{\pm 0.2}$ | $67.6_{\pm 0.2}$ | $66.7_{\pm 0.3}$ | $67.2_{\pm 0.2}$ |
| BS (Ren et al., 2020) | $61.1_{\pm 0.2}$ | $48.5_{\pm 0.2}$ | $31.8_{\pm 0.4}$ | $50.9_{\pm 0.1}$ | $65.5_{\pm 0.2}$ | $67.5_{\pm 0.1}$ | $67.5_{\pm 0.1}$ | $67.2_{\pm 0.2}$ |
| BS + CUDA | $61.9_{\pm 0.1}$ | $49.2_{\pm 0.0}$ | $32.3_{\pm 0.4}$ | $51.6_{\pm 0.1}$ | $67.6_{\pm 0.1}$ | $68.3_{\pm 0.1}$ | $68.3_{\pm 0.1}$ | $68.2_{\pm 0.1}$ |
| RIDE (3 experts) (Wang et al., 2021)⋆ | $64.8_{\pm 0.1}$ | $50.8_{\pm 0.2}$ | $34.6_{\pm 0.2}$ | $53.6_{\pm 0.1}$ | $70.4_{\pm 0.1}$ | $71.8_{\pm 0.1}$ | $71.7_{\pm 0.1}$ | $71.6_{\pm 0.1}$ |
| RIDE + CMO (Park et al., 2022)⋆ | 65.6 | 50.6 | 34.8 | 54.0 | 68.0 | 70.6 | 72.0 | 70.9 |
| RIDE (3 experts) + CUDA⋆ | $65.9_{\pm 0.1}$ | $51.7_{\pm 0.2}$ | $34.9_{\pm 0.2}$ | $54.7_{\pm 0.1}$ | $70.7_{\pm 0.2}$ | $72.5_{\pm 0.1}$ | $72.7_{\pm 0.2}$ | $72.4_{\pm 0.2}$ |
| BCL (Zhu et al., 2022) | $65.3_{\pm 0.2}$ | $53.5_{\pm 0.2}$ | $36.3_{\pm 0.3}$ | $55.6_{\pm 0.2}$ | $69.5_{\pm 0.1}$ | $72.4_{\pm 0.2}$ | $71.7_{\pm 0.1}$ | $71.8_{\pm 0.1}$ |
| BCL + CUDA | $66.8_{\pm 0.1}$ | $\mathbf{53.9}_{\pm 0.3}$ | $\mathbf{36.6}_{\pm 0.2}$ | $56.3_{\pm 0.1}$ | $70.9_{\pm 0.2}$ | $\mathbf{72.8}_{\pm 0.1}$ | $72.0_{\pm 0.1}$ | $72.3_{\pm 0.1}$ |

Table 3: Comparison for CIFAR-LT-100 performance on ResNet-32 with 400 epochs.

| Algorithm | Imbalance Ratio | |
|---|---|---|
| | 100 | 50 |
| PaCo | 52.0 | 56.0 |
| BCL | 52.6 | 57.2 |
| NCL | 54.2 | 58.2 |
| BCL + CUDA | 53.5 | 57.4 |
| NCL + CUDA | **54.8** | **59.6** |

Table 4: Augmentation analysis on CIFAR-100-LT with IR 100. AA (Cubuk et al., 2019), FAA (Lim et al., 2019), DADA (Li et al., 2020b), and RA (Cubuk et al., 2020) with $n = 1, m = 2$ policies are used. C, S, I represent CIFAR, SVHN, and ImageNet policy.

| | Vanilla | AA | | | FAA | | DADA | | RA | CUDA |
|---|---|---|---|---|---|---|---|---|---|---|
| | | C | S | I | C | I | C | I | | |
| CE | 38.7 | 41.7 | 40.7 | 40.1 | 42.3 | 40.8 | 41.0 | 41.2 | 40.5 | **42.7** |
| CE-DRW | 41.4 | 46.5 | 44.7 | 45.5 | 46.3 | 44.8 | 45.6 | 45.7 | 45.8 | **47.4** |
| LDAM-DRW | 42.5 | 47.0 | 44.7 | 44.9 | 46.6 | 45.6 | 45.9 | 46.5 | 44.0 | **47.2** |
| BS | 43.3 | 47.0 | 46.1 | 45.5 | 46.5 | 45.0 | 45.0 | 46.9 | 45.2 | **47.7** |
| RIDE (3 experts) | 49.7 | 49.5 | 47.3 | 45.5 | 49.8 | 50.6 | 50.4 | 50.5 | 47.9 | **50.7** |

CUDA, we apply a $p_{\text{aug}}$ of 0.5 and $T$ of 10 for all experiments. For $\gamma$, we set the values as 0.6 for CIFAR-100-LT and 0.4 for ImageNet-LT and iNaturalist 2018. The detailed implementation for baselines are in Appendix B.

## 4.2 EXPERIMENTAL RESULTS

In this section, we report the performances of the methods compared on the CIFAR-100-LT, ImageNet-LT, and iNaturalist 2018. We include four different categories of accuracy: all, many, med(ium), and few. Each represents the average accuracy of all samples, classes containing more than 100 samples, 20 to 100 samples, and under 20 samples, respectively.

**CIFAR-100-LT.** In Table 1, we report the performance when CUDA is applied to the various algorithms: CE, CE-DRW (Cao et al., 2019), LDAM-DRW (Cao et al., 2019), BS (Ren et al., 2020), RIDE (Wang et al., 2021) with 3 experts, RIDE+CMO (Park et al., 2022), and BCL (Zhu et al., 2022). Compared to the cases without CUDA, balanced validation performance is increased when we apply the proposed approach.

Recently, some works (Cui et al., 2021; Alshammari et al., 2022; Zhu et al., 2022; Li et al., 2022a) have shown impressive performances with diverse augmentation strategies and longer training epochs. For a fair comparison with these methods, we examine CUDA using the same experimental setups from PaCo (Cui et al. 2021; 400 epochs with batch size of 64). Table 3 shows that augmented images using CUDA can enhance LTR performance compared to the other baselines. In particular, CUDA with NCL obtains the best performance over 400 epochs. As noted by Li et al. (2022a), the NCL algorithm utilizes six times as much memory compared to the vanilla architecture with three experts. Hereinafter in large-scale benchmarks, we focus on the cases with similar network size.

**ImageNet-LT and iNaturalist 2018.** To evaluate the performance of CUDA on larger datasets, we conduct experiments on ImageNet-LT (Liu et al., 2019) and iNaturalist 2018 (Van Horn et al., 2018). Table 2 summarizes the performance of various LTR methods and the performance gain when integrated with CUDA. Our proposed method consistently improves performance regardless of the LTR method and target dataset by simply adding class-wise data augmentation without complicated methodological modification. Additionally, to evaluate the performance gain of CUDA on other

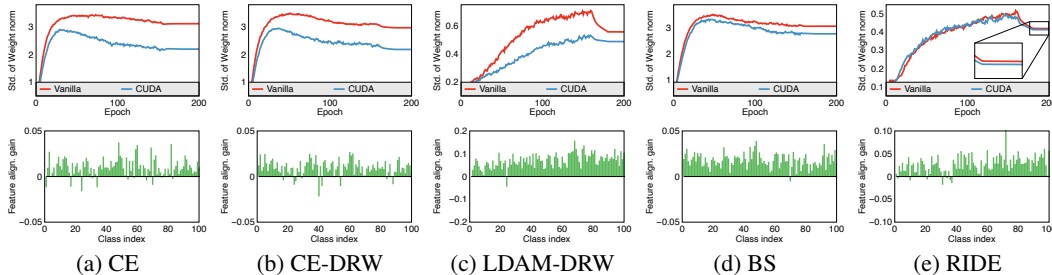

(a) CE      (b) CE-DRW      (c) LDAM-DRW      (d) BS      (e) RIDE

Figure 3: Analysis of how CUDA improves long-tailed recognition performance, classifier weight norm (top row) and feature alignment gain (bottom row) of the CIFAR-100-LT validation set. Notably that weight norm and feature alignment represent class-wise weight magnitude of classifier and ability of feature extractor, respectively. The detailed analysis is described in Section 4.3.

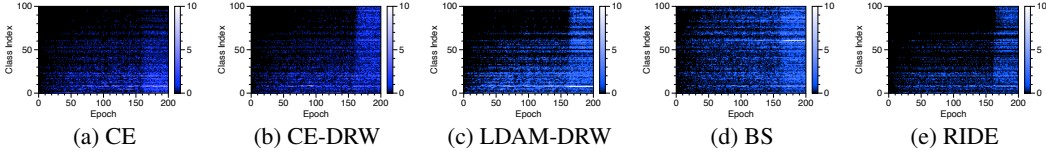

(a) CE      (b) CE-DRW      (c) LDAM-DRW      (d) BS      (e) RIDE

Figure 4: Evolution of LoL score on various algorithms, CE, CE-DRW, LDAM-DRW, BS, and RIDE.

architectures, we experiment with CUDA on ImageNet-LT with ResNet-10 (Liu et al., 2019) and ResNeXt-50 (Xie et al., 2017), as reported in Appendix C.

## 4.3 ANALYSIS

We design our analyses to answer the following questions. (1) How does CUDA perform? (2) Does CUDA perform better than other augmentation methods? (3) How does LoL score change over training epochs when combined with various LTR methods? (4) Which part of CUDA is important to improved performance? These analyses provide additional explanations to understand CUDA. All experiments are conducted on CIFAR-100-LT with imbalance ratio of 100.

**How does CUDA mitigate the class imbalance problem?** To deeply understand CUDA, we observe two types of metrics: (1) variance of weight L1-Norm of linear classifier between each class (2) feature alignment gain for each class (*i.e.,* cosine similarity with and without CUDA) on validation dataset. The classifier weight norm is usually used to measure how balanced the model consider the input from a class-wise perspective (Kang et al., 2020; Alshammari et al., 2022). Feature alignment, especially feature cosine similarity amongst samples belonging to the same class, is a measure of the extent to which the extracted features are aligned (Oh et al., 2021). As shown in Figure 3, CUDA has two forces for alleviating imbalance. For all cases, CUDA reduces the variance of the weight norm (*i.e.,* balance the weight norm), and thus the trained model consider the minority classes in a balanced manner. Note that because LDAM-DRW and RIDE utilize a cosine classifier (*i.e.,* utilizing L2 normalized linear weight), their standard deviation scale is quite different from those other methods. Because LDAM-DRW, BS, and RIDE include balancing logic in their loss function, they exhibit lower variance reduction compared to the CE and CE-DRW. Second, as shown in the bottom row in Figure 3, CUDA obtains feature alignment gains for almost all classes. This shows that CUDA facilitates a network to learn to extract meaningful features.

**Compared with other augmentations.** To verify the impact of CUDA, we examine the other augmentation methods as follows. We compare five augmentation methods, including AutoAugment (AA, Cubuk et al. 2019), Fast AutoAugment (FAA, Lim et al. 2019), DADA (Li et al., 2020b), RandAugment (RA, Cubuk et al. 2020), and the proposed method CUDA. Because AA, FAA, and DADA provide their policies searched by using CIFAR, SVHN (for AA), and ImageNet, we leverage their results. Furthermore, RA suggests using their parameter $(n, m) = (1, 2)$ for CIFAR, and we follow their guidelines. As shown in Table 4, even though the automated augmentation methods use additional computation resources to search, CUDA outperforms the other pre-searched augmentations. This shows that CUDA is computationally efficient.

**Dynamics of LoL score.** We evaluate how LoL scores vary with algorithms: CE, CE-DRW, LDAM-DRW, BS, and RIDE. Note that we set a lower class index (*i.e.,* 0) as the most common class (*i.e.,* the

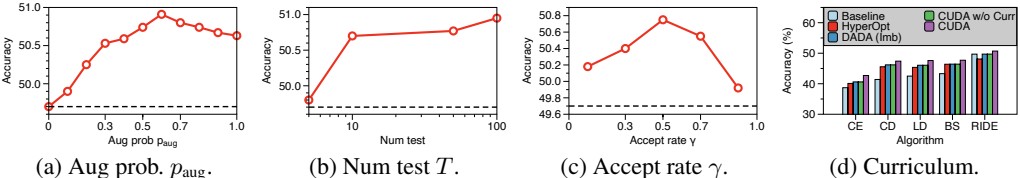

(a) Aug prob. $p_{aug}$.  (b) Num test $T$.  (c) Accept rate $\gamma$.  (d) Curriculum.

Figure 5: Additional analysis of CUDA. (a) sensitivity of augmentation probability $p_{aug}$, (b) sensitivity analysis of number of sample coefficient $T$, (c) sensitivity of acceptance threshold $\gamma$, and (d) impact of curriculum. The dotted lines in (a), (b) and (c) represents the performance of CE.

number of samples is 500), while an index of 100 represents the rarest class (*i.e.,* with five samples). As described in Figure 4, as training progressed, the LoL score of all algorithms increase. After learning rate decay (*i.e.,* 160 epoch) all algorithms are able to learn to classify minority classes more easily than before. In particular, except for BS, the majority classes of most algorithms show a steep increment. The reason that BS exhibit a similar increasing speed for majority and minority classes is that it includes a module to balance the impact of majority and minority samples. Furthermore, we found that CE-DRW and BS have similar end average accuracy in the case of applying CUDA but different LoL score dynamics. We can conclude that LoL score on one category of classes has a high correlation with the performance of opposite classes from the observation that CE-DRW has higher and lower performance gain for many and few, respectively, than BS.

**Parameter sensitivity.** For further analysis, we conduct a sensitivity analysis of hyperparameters in CUDA. More precisely, we study three kinds of parameters, including augmentation probability $p_{aug}$ (Figure 5a), number of tests $T$ (Figure 5b), and LoL update threshold $\gamma$ (Figure 5c). We examine each hyperparameter sensitivity on a CUDA case with RIDE and the remainder of the hyperparameters are fixed to the default values in Section 4.1. All results show that the performance gains of CUDA decreases if the parameters are adjusted to make the augmentation too strong or weak. For example, the augmentation strength of all classes steeply increases when $\gamma$ becomes small. The strength cannot increase when $\gamma$ becomes large, and thus it cannot improve the performance of the model. Moreover, as shown in Figure 5b, the performance of CUDA increases as $T$ increases. However, larger $T$ spends computational overhead, we set $T$ as 10 and obtained cost-effective performance gain.

**Impact of curriculum.** In addition to studying the impact of CUDA, we examine its performance component-wise. In particular, we test the case where class-wise augmentation strength is searched based on the hyperparameter optimization algorithm. We check five cases overall: baseline algorithm, hyperparameter optimization (HO), re-searched DADA for CIFAR-100-LT, CUDA without curriculum, (*i.e.,* re-training utilizing the final augmentation strength of CUDA), and CUDA. We provide detailed description for each method in Appendix E. As described in Figure 5d, CUDA finds better augmentation strengths compare to the hyperparameter search case. This means that CUDA exhibits not only a lower searching time but also obtains better augmentation strength. Moreover, by comparing the performance of with or without curriculum, the curriculum also can provide additional advance to the model to achieve better generalization. Additionally, as Figure 4, lower augmentation strength at the beginning of training is more effective than static higher augmentation strength. These results are consistent with the results of previous studies on curriculum learning methods (Zhou et al., 2020b).

## 5 CONCLUSION

In this study, we proposed CUDA to address the class imbalance problem. The proposed approach is also compatible with existing methods. To design a proper augmentation for LTR, we first studied the impact of augmentation strength for LTR. We found that the strength of augmentation for a specific type of class (*e.g.,* major class) could affect the performance of the other type (*e.g.,* minor class). From this finding, we designed CUDA to adaptively find an appropriate augmentation strength without any further searching phase by measuring the LoL score for each epoch and determining the augmentation accordingly. To verify the superior performance of proposed approach, we examined each performance with various methods and obtained the best performance among the methods compared, including synthetically generated and real-world benchmarks. Furthermore, from our analyses, we validated that our CUDA enhanced balance and feature extraction ability, which can consistently improve performance for majority and minority classes.

## ACKNOWLEDGEMENT

This work was supported by Institute of Information & communications Technology Planning & Evaluation (IITP) grant funded by the Korea government (MSIT) (No.2019-0-00075, Artificial Intelligence Graduate School Program (KAIST), 10%) and the Institute of Information & communications Technology Planning & Evaluation (IITP) grant funded by the Korea government (MSIT) (No. 2022-0-00871, Development of AI Autonomy and Knowledge Enhancement for AI Agent Collaboration, 90%)

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

# Appendix
## CUDA: Curriculum of Data Augmentation for Long-tailed Recognition

Owing to the page limitation of the main manuscript, we provide detailed information in this supplementary as follows. (1) In Appendix A, we summarize the experimental setup of Figure 1, and further explain why augmentation on one side causes performance degradation on the opposite side. (2) In Appendix B, we describe in detail our experimental setting, including dataset configuration, data preprocessing, and training implementation. (3) In Appendix C, we show ImageNet-LT performance on different size and architecture networks, training time analysis, and accuracy on the balanced dataset case. (4) In Appendix D, we present in detail the augmentation operations that CUDA utilizes. (5) In Appendix E, we describe the experimental setting of Figure 5d.

## A  DETAIL FOR FIGURE 1

### A.1  EXPERIMENTAL SETTINGS

**Major and minor group decomposition.**  To check the impact of augmentation on majority and minority classes, we split the training dataset into two clusters. The majority cluster is the top $50$ classes by sorting through the number of samples for each class. The bottom $50$ classes are in the minority cluster. For simplicity, we utilize class indices of $0$ to $49$ as the majority and $50$ to $99$ as the minority, respectively. For the balanced case, we utilize $0$ to $49$ classes as cluster 1, and the others as cluster 2.

**Controlling augmentation strength.**  We set the augmentation strength as the number of augmentation and its augmentation magnitude by following the augmentation rule of CUDA. For example, the samples in the majority classes with magnitude parameter $4$ represents that they are augmented with randomly sampled $4$ augmentations with their own pre-defined augmentation magnitude.

**Training setting.**  For heatmaps in Figure 1, we follow the training recipe of CIFAR-100-LT for CE case, *e.g.,* ResNet-32, learning rate of $0.1$, and so on. Further details, hyperparameters, and datasets are described in section 4 and Appendix B.

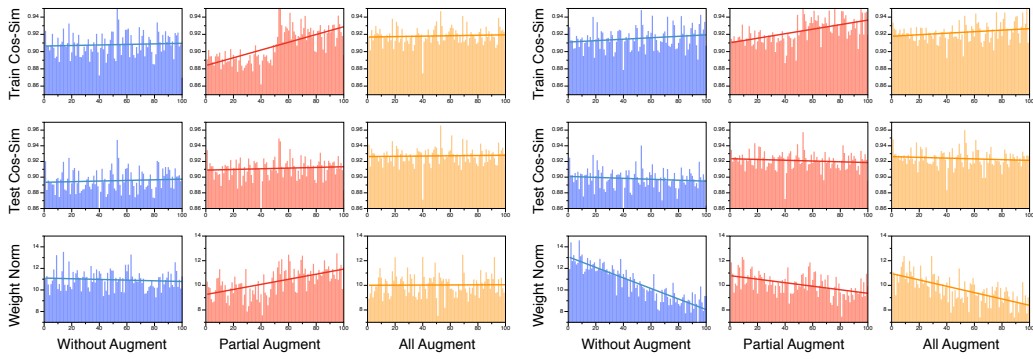

Figure 6: Analysis on Balanced CIFAR-100.   Figure 7: Analysis on CIFAR-100-LT (IR $100$).

### A.2  ANALYSIS

**Analysis for Figure 1.**  To figure out the reason for the phenomena in Figure 1, we conduct further analysis as shown in Figure 6 and Figure 7. Our experimental setups are as follows:

- Train the networks with three augmentation strategies, respectively (without, partial, and all), then measure the class-wise feature alignment and linear classifier weight norm for all networks. (Experiment 1)

- From a trained network without augmentation in Experiment 1, we freeze the feature extractor and train the linear classifier layer with augmenting partial classes. Then, we measure the class-wise L1-norm for all linear classifiers. (Experiment 2)

From the Figure 6 and Figure 7, we have three observations from Experiment 1:

1. When we conduct augmentation only for partial classes (0-49 classes), the feature alignment for augmented classes of the training dataset is degraded compared to the non-augmented classes. This is because the augmentation classes have more diversified training data than non-augmentation classes, which leads to more diversification in feature space. We observe the balance between alignment between classes in the cases of without augmentation and with all augmentation since all classes have similar diversity. (See the first rows in Figure 6, 7)

2. However, all three augmentation strategies have balanced class-wise feature alignment for the same test dataset. This tendency can be observed in both balanced and imbalanced datasets. This result is consistent with Kang et al. (2020). Furthermore, the values for feature alignment are increased when we conduct augmentation partially or all, compared to without augmentation. This result shows that augmentation enhances the feature extraction ability, which is consistent with conventional studies. (See the second rows in Figure 6, 7)

3. When we conduct augmentation only for partial classes on a balanced dataset, the class-wise weight norm of the linear classifier is larger for non-augmentation classes. This result incurs performance improvement for non-augmentation classes and reduction for augmentation classes since this linear classifier has a tendency to classify non-augmented classes with larger weight values. However, we observe that class-wise weight norms are balanced in "without augmentation" and "all augmentation" cases. (See the third row in Figure 6)

4. We observe that the class-wise weight norm of the linear classifier is larger for majorities for all classes that have the same augmentation strength. These results are consistent with previous works (Kang et al., 2020; Alshammari et al., 2022). However, when we conduct augmentation only for majorities, the class-wise weight norm is more balanced. This phenomenon is similar to the balanced case in that partial augmentation incurs a reduction in the norm of the linear classifier for augmented classes. (See the third row in Figure 7)

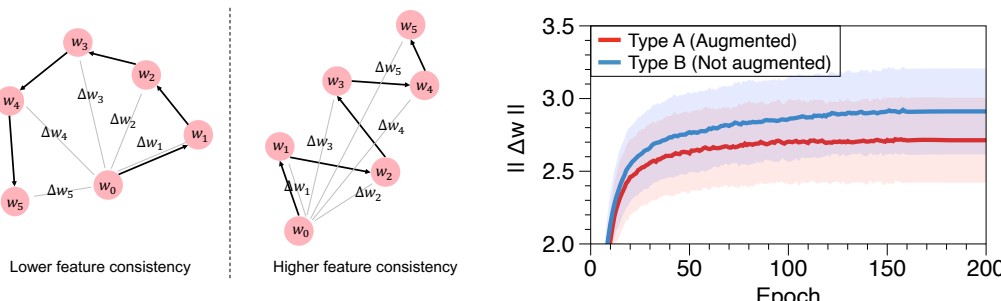

Figure 8: Concept of the impact of lower feature alignment on linear classifier.

Figure 9: The difference of linear classifier norm $\|\Delta \mathbf{w}\|$ along training epoch.

Our observations from Experiment 1 are highly consistent in both balanced and imbalanced datasets. The results in Figure 1, Figure 6 and Figure 7 highly motivate the design of CUDA. Moreover, our results for Experiment 2 can explain these observations as shown in Figure 8 and Figure 9.

We observe that in the presence of feature alignment degradation from augmentation, the corresponding norm is relatively small, as shown in Figure 8. This is because in the class that has lower feature alignment, the variation of the gradient for the linear classifier is larger than in the class with high feature alignment. As shown in Figure 9, from Experiment 2, we observe that $\|\Delta \mathbf{w}\|$, the norm of class-wise difference of between current and initialize linear classifier parameters $\Delta \mathbf{w} := \mathbf{w} - \mathbf{w}_0$, have smaller value in augmented classes than non-augmented classes. From our experimental analysis in Figure 6, 7, and 9, we can conclude that augmentation breaks the consistency of feature alignment and it makes the weight norm of the linear classifier decreases.

# B    IMPLEMENTATION DETAIL IN SECTION 4

## B.1    DATASET DESCRIPTION

**CIFAR-100-LT.** CIFAR-100-LT is a subset of CIFAR-100. Following Wang et al. (2021); Park et al. (2022); Zhu et al. (2022), we use the same long-tailed version for a fair comparison. The number of samples of $k$th class is determined as follows: (1) Compute the imbalanced factor $N_{max}/N_{min}$, which reflects the degree of imbalance in the data. (2) $|\mathcal{D}_k|$ between $|\mathcal{D}_1| = N_{max}$ and $|\mathcal{D}_{100}| = N_{min}$ follows an exponential decay (*i.e.,* $|\mathcal{D}_k| = |\mathcal{D}_1| \times (N_{max}/N_{min})^{k/100}$). The imbalance factors used in the experiment are set to 100, 50, and 10.

**ImageNet-LT.** ImageNet-LT (Liu et al., 2019) is a modified version of the large-scale real-world dataset (Russakovsky et al., 2015). Subsampling is conducted by following the Pareto distribution with power value $\alpha = 0.6$. It consists of 115.8K images of $1,000$ classes in total. The most common or rare class has $1,280$ or $5$ images, respectively.

**iNaturalist 2018.** iNaturalist (Van Horn et al., 2018) is a large-scale real-world dataset which consists of 437.5K images from $8,142$ classes. It has long-tailed property by nature, with an extremely class imbalanced. In addition to long-tailed recognition, this dataset is also used for evaluating the fine-grained classification task.

## B.2    DATA PREPROCESSING

For data preprocessing, we follow the default settings of Cao et al. (2019). For CIFAR-100-LT, each side of the image is padded with 4 pixels, and a $32 \times 32$ crop is randomly selected from the padded image or its horizontal flip. For ImageNet-LT and iNaturalist 2018, after resizing each image by setting the shorter side to 256 pixels, a $224 \times 224$ crop is randomly sampled from an image or its horizontal flip.

For BCL and NCL, which use AutoAugment (Cubuk et al., 2019) or RandAugment (Cubuk et al., 2020) as default data augmentation, we apply them after random cropping by following their original papers (Zhu et al., 2022; Li et al., 2022a). Then, we finally conduct CUDA after all default augmentation operations, and then normalize the image with following mean and standard deviation values sequentially: CIFAR-100-LT ((0.4914, 0.4822, 0.4465), (0.2023, 0.1994, 0.2010)), ImageNet-LT ((0.485, 0.456, 0.406), (0.229, 0.224, 0.225)), and iNaturalist 2019 ((0.466, 0.471, 0.380), (0.195, 0.194, 0.192)).

## B.3    DETAILED IMPLEMENTATION

Because some official codes do not open their entire implementations, we re-implement by following the rules. For re-implementation, we reproduce the code based on their partial code and the authors' responses.

**RIDE.** We follow the officially offered code[2]. Among various experimental configurations of official code (*e.g.,* one-stage RIDE, RIDE-EA, Distill-RIDE), for fair comparison (to leverage similar computation resources), we utilize one-stage training (*i.e.,* one-stage RIDE) for all cases. We confirm that CMO (Park et al., 2022) also utilizes this setup for RIDE + CMO from the response of the authors.

**CMO.** We re-implement all CMO results from their official code[3] in our work. However, the official code of CMO does not contain code for RIDE + CMO. Therefore, we re-implement by injecting the CMO part for BS in the official code (weighted sampler and mixup part) into the RIDE code. Furthermore, for iNaturalist 2018, we train the model for 100 epochs for a fair comparison with other methods (whereas the original RIDE + CMO is trained for 200 epochs on iNaturalist 2018).

**BCL.** The officially released code[4] of BCL only contains ImageNet-LT and iNaturalist 2018. Whereas the official code applies a cosine classifier for ImageNet-LT and iNaturalist 2018, we apply

---

[2]https://github.com/frank-xwang/RIDE-LongTailRecognition
[3]https://github.com/naver-ai/cmo
[4]https://github.com/FlamieZhu/Balanced-Contrastive-Learning

an ordinary linear classifier for CIFAR-100-LT from the author's response. All hyperparameters are the same as the experiment settings of the original work (Zhu et al., 2022).

### B.4    GUIDELINE FOR HYPER-PARAMETER TUNING

Although we did not tune the hyper-parameters extensively. However, we give a guideline to select the hyper-parameters.

**The number of samples for updating LoL** ($T$). We can set this value according to the given computing resources (*i.e.,* the largest $T$ under computing resource constraint). This is because the performance improves as $T$ increases from obtaining a definite LoL score by testing many samples.

**The acceptance threshold** ($\gamma$). Our strategy for tuning gamma is to select the largest value in which at least one of LoL scores among all classes increases within 20 epochs. This is because for large-scale datasets, the network fail to infer even the easier-to-learn majority classes. Here is the detailed tuning strategy for $\gamma$.

- We initially set $\gamma$ as 0.6.
- We decrease the threshold $\gamma$ by 0.1 points whenever it fails to raise any of LoL score for the first 20 training epochs.

We condcut this search on CE with CIFAR-100-LT with IR 100 and using the same $\gamma$ value of the other algorithms with remaining IR settings. Also, we conduct this search rule on ImageNet-LT with CE and use the same value to the other large-scale dataset, *i.e.,* iNaturalist 2018 with remaining algorithms.

**The augmentation probability** ($p_{\text{aug}}$). While we did not tune this hyper-parameter, we offer the guideline how to tune this value based on Figure 5a. As shown in Figure 5a, the shape of graph between $p_{\text{aug}}$ and performance is concave. Thanks to concavity, we think that it is easy to find the optimal value for this hyper-parameter. Note that the reason for the concavity is because the decision of $p_{\text{aug}}$ value has a trade-off between preserving the information of the original image and exploring diversified images.

**Further sensitivity analysis on ImageNet-LT** In Section 4, we apply different values of $\gamma$ in CIFAR-100-LT (0.6) and large-scale datasets (0.4; ImageNet-LT and iNaturalist 2018). In addition to Figure 5, we further conduct the sensitivity analysis for $\gamma$ on the ImageNet-LT to verify CUDA works well robustly with different values of $\gamma$ on large-scale datasets. As shown in Table 5, our proposed method CUDA is also robust to hyper-parameter selection for $\gamma$ not only the small datasets such as CIFAR-100-LT but also large-scale datasets.

Table 5: Sensitivity analysis of $\gamma$ with BS on ImageNet-LT dataset

| $\gamma$ | 0.3 | 0.4 | 0.5 | 0.6 |
|---|---|---|---|---|
| Acc. (%) | 51.42 | 51.59 | 51.38 | 51.24 |

## C    FURTHER ANALYSES

**Training Time Analysis.** CUDA requires additional computation for computing LoL score. We measure the additional training time for adding CUDA on various algorithms. As shown in Figure 11, when utilizing CUDA additional training time is spent. However, the additional operation for searching the LoL score does not require a large value. For example, BS with CUDA spends $\times 1.29$ time to obtain adequate augmentation strength.

**Network Architecture Analysis.** We also present our ResNet-10 (Liu et al., 2019) and ResNeXt-50 (Xie et al., 2017) experiments on the ImageNet-LT dataset in Figure 10, respectively. These results show that CUDA consistently improves performance regardless of network sizes and corresponding LTR methods.

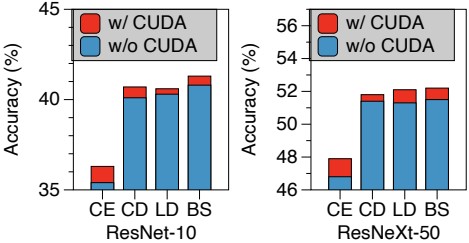

Figure 10: Network architecture.

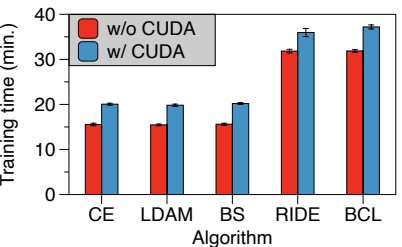

Figure 11: Training time.

**What if `CUDA` is ran on the balanced dataset.** We examine that if CUDA is applied to the balanced case, *i.e.,* imbalance ratio is 1. As described in the Table 6 CUDA obtains 1.9% accuracy gain, which is lower than the other auto augmentation methods. However, other autoaugmentation methods spend more computation time searching a good augmentation than CUDA. Furthermore, as described in Figure 4, CUDA has higher performance than the others when the class imbalance dataset is given.

Table 6: Balanced case. † mark represents the reported value in (Li et al., 2020b)

| Augmentation | Acc. | Searching time (Overhead) |
|---|---|---|
| CE | 68.5 | - |
| AutoAug | 70.7 | 5, 000 GPU hours† |
| RandAug | 69.4 | - |
| FAA | 70.7 | 3.5 GPU hours† |
| DADA | 70.9 | 0.2 GPU hours† |
| CUDA | 70.4 | 0.07 GPU hours |

# D    AUGMENTATION PRESET

## D.1    DATA AUGMENTATION OPERATIONS USED IN `CUDA`.

There have been numerous data augmentation operations in vision tasks. We used totally 22 augmentations for CUDA with their own parameter set. Details of the operation set and parameters are described in Table 7. For augmentation magnitude parameter $m_k(s)$, we divide parameters into thirty values linearly. For example of, ShearX case, its max and min values are 3 and 0, respectively. Therefore, $m_{\text{ShearX}}(s) = (3 - 0)/30 * s$, thus $m_{\text{ShearX}}(1) = 0.01 = (3 - 0)/30 * 1$.

## D.2    FURTHER ANALYSIS ON AUGMENTATION PRESET

To get further intuition on the effect of number of predefined augmentation operations, we conduct several exploratory experiments.

**Validity of our main finding (Figure 1) under a few predefined augmentation.** The observation in Figure 1 is caused by minorities becoming relatively easy to learn since majorities have become difficult. Therefore, if the sample of majorities becomes difficult enough to learn, the same phenomenon as Figure 1 occurs regardless of the number of augmentation presets. To verify that our main finding is valid regardless of the number of predefined augmentations, we conduct the experimental with ten augmentation operations (Mirror, ShearX, Invert, Smooth, ResizeCrop, Color, Brightness, Sharpness, Rotate, AutoContrast). Table 8 describes the performance of (0,0), (0,4), (4,0), and (4,4) that each configuration denotes the augmentation strength of (majority; top 50 class, minor; bottom 50 class). Through the results, we verify that the finding in Figure 1 is valid even in a small number of predefined augmentation operations.

**Effect of number of predefined augmentation.** We further analyze the impact of predefined augmentation operations ($K$ in Figure 2); we additionally experiment by replacing the augmentation preset in Appendix D with the following two augmentation presets: (1) 10 randomly sampled augmentations (Mirror, ShearX, Invert, Smooth, ResizeCrop, Color, Brightness, Sharpness, Rotate,

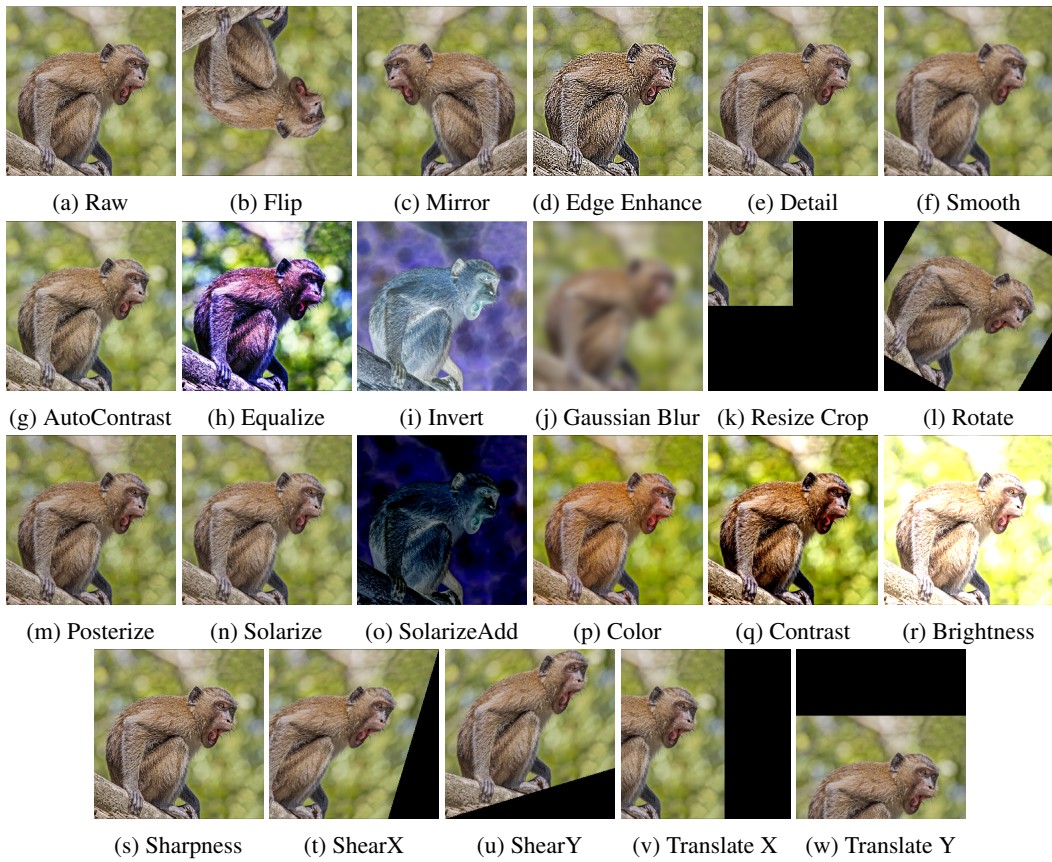

(a) Raw (b) Flip (c) Mirror (d) Edge Enhance (e) Detail (f) Smooth

(g) AutoContrast (h) Equalize (i) Invert (j) Gaussian Blur (k) Resize Crop (l) Rotate

(m) Posterize (n) Solarize (o) SolarizeAdd (p) Color (q) Contrast (r) Brightness

(s) Sharpness (t) ShearX (u) ShearY (v) Translate X (w) Translate Y

Table 7: Description of augmentation operations utilized in `CUDA`. We show the examples of each augmentation with maximum augmentation parameters.

| Operation | Parameter | Description |
|---|---|---|
| Flip | On/Off | Flip top and bottom |
| Mirror | On/Off | Flip left and right |
| Edge Enhancement | On/Off | Increasing the contrast of the pixels around the targeted edges |
| Detail | On/Off | Utilize convolutional kernel $[[0, -1, 0], [-1, 10, -1], [0, -1, 0]]$ |
| Smooth | On/Off | Utilize convolutional kernel $[[1, 1, 1], [1, 5, 1], [1, 1, 1]]$ |
| AutoContrast | On/Off | Remove a specific percent of the lightest and darkest pixels |
| Equalize | On/Off | apply non-linear mapping to make uniform distribution |
| Invert | On/Off | Negate the image |
| Gaussian Blur | [0,2] | Blurring an image using Gaussian function |
| Resize Crop | [1,1.3] | Resizing and center random cropping |
| Rotate | [0,30] | Rotate the image |
| Posterize | [0,4] | Reduce the number of bits for each channel |
| Solarize | [0,256] | Invert all pixel values above a threshold |
| SolarizeAdd | [0,110] | Adding value and run solarize |
| Color | [0.1, 1.9] | Colorize gray scale values |
| Contrast | [0.1,1.9] | Distance between the colors |
| Brightness | [0.1,1.9] | Adjust image brightness |
| Sharpness | [0.1,1.9] | Adjust image sharp |
| Shear X | [0,0.3] | Shearing X-axis |
| Shear Y | [0,0.3] | Shearing Y-axis |
| Translate X | [0,100] | Shift X-axis |
| Translate Y | [0,100] | Shifting Y-axis |

AutoContrast) and (2) RandAugment (Cubuk et al., 2020) preset that consists of (AutoContrast,

Table 8: The performance comparison between different augmentation strength on major (class indices 0-49) and minor (class indices 50-99) categories.

| | (major, minor) | Many | Med | Few | All | | (major, minor) | Many | Med | Few | All |
|---|---|---|---|---|---|---|---|---|---|---|---|
| CE | 0,0 | 66.2 | 37.3 | 8.2 | 38.7 | CE-DRW | 0,0 | 62.8 | 41.7 | 16.2 | 41.4 |
| | 0,4 | 69.7 | 30.4 | 2.3 | 35.7 | | 0,4 | 65.9 | 37.2 | 10.6 | 39.3 |
| | 4,0 | 60.9 | 39.3 | 12.8 | 38.9 | | 4,0 | 49.3 | 45.2 | 28.3 | 41.6 |
| | 4,4 | 67.0 | 34.5 | 4.7 | 37.0 | | 4,4 | 56.6 | 46.6 | 24.6 | 43.5 |
| LDAM-DRW | 0,0 | 62.8 | 42.3 | 19.0 | 42.5 | BS | 0,0 | 61.6 | 42.3 | 23.0 | 43.3 |
| | 0,4 | 70.1 | 34.3 | 6.4 | 38.5 | | 0,4 | 66.9 | 37.9 | 10.8 | 39.9 |
| | 4,0 | 52.3 | 42.0 | 27.7 | 41.3 | | 4,0 | 48.7 | 42.5 | 28.7 | 40.5 |
| | 4,4 | 61.1 | 43.4 | 17.3 | 41.8 | | 4,4 | 56.3 | 44.2 | 23.0 | 42.1 |
| RIDE | 0,0 | 67.7 | 51.5 | 26.7 | 49.7 | | | | | | |
| | 0,4 | 70.5 | 36.8 | 7.7 | 39.9 | | | | | | |
| | 4,0 | 56.6 | 44.5 | 27.2 | 43.6 | | | | | | |
| | 4,4 | 62.3 | 44.5 | 21.6 | 43.9 | | | | | | |

Equalize, Invert, Rotate, Posterize, Solarize, SolarizeAdd, Color, Contrast, Brightness, Sharpness, ShearX, ShearY, CutoutAbs, TranslateXabs, TranslateYabs). Table 9 demonstrates that the accuracy slightly increases when the size of the augmentation preset increases. However, the gap between the RandAugment preset (14 operations) and our original preset (22 operations) is small compared to the gap between the vanilla (without CUDA case) and the RandAugment case. These results verify our belief that the impact of the number of predefined augmentations is small.

Table 9: The performance comparison of CUDA with different number ($K$) of predefined augmentation operations.

| | Category | CE | CE-DRW | LDAM-DRW | BS | RIDE |
|---|---|---|---|---|---|---|
| Vanilla (w/o augmentation) | Many | 66.2 | 62.8 | 62.8 | 61.6 | 67.7 |
| | Med | 37.3 | 41.7 | 42.3 | 42.3 | 51.5 |
| | Few | 8.2 | 16.2 | 19.0 | 23.0 | 26.7 |
| | All | 38.7 | 41.4 | 42.5 | 43.3 | 49.7 |
| Random Selection (K=10) | Many | 70.8 | 62.3 | 65.2 | 62.6 | 68.5 |
| | Med | 40.4 | 49.0 | 49.2 | 46.9 | 52.0 |
| | Few | 9.0 | 26.7 | 21.6 | 27.3 | 27.1 |
| | All | 41.6 | 47.0 | 46.5 | 46.5 | 50.3 |
| RandAugment (K=14) | Many | 70.3 | 63.5 | 65.4 | 62.9 | 68.5 |
| | Med | 40.7 | 49.1 | 50.6 | 48.1 | 52.3 |
| | Few | 9.6 | 26.0 | 21.6 | 28.7 | 27.0 |
| | All | 41.8 | 47.2 | 47.1 | 47.5 | 50.4 |
| Ours (K=22) | Many | 71.6 | 64.3 | 67.3 | 63.3 | 69.2 |
| | Med | 42.3 | 49.2 | 50.4 | 48.4 | 52.8 |
| | Few | 9.4 | 26.7 | 21.4 | 28.7 | 27.3 |
| | All | 42.7 | 47.7 | 47.6 | 47.7 | 50.7 |

**Effect of randomly ordered data augmentation.** Our proposed CUDA operates randomly sequential of the selected augmentations based on the strength of DA. To study the impact of these randomly ordered augmentations, we compare CUDA and CUDA with fixed order augmentations. For examples, when the operation indices $(6, 3, 5)$ among 22 augmentations are samples, it is applied with $(3, 5, 6)$. Table 10 shows small performance differences between the two methods. Thus, we believe that the effect of the augmentation order on the difficulty is negligible. This is because the effectiveness of CUDA is expected to be sufficiently high even in a given order of augmentations since the goal is to make it harder to learn, regardless of the ordered (determined or random) order.

**Comparison with random augmentation.** To verify that the success of CUDA is not simply from a richer dataset made by DA, we compare our proposed method CUDA to randomly sampled augmentation for every iteration. Our comparison methods are Random 5 and Random 10, which denote the conduct of five and ten randomly sampled augmentations for every iteration. As shown in Table 11, while Random 10 generates the most diversifying images, the network trained with this showed the worst performance, even lower than vanilla. Our CUDA achieves the best performance among all methods.

Table 10: The performance comparison of `CUDA` with random order (Ours) and fixed order of augmentation operations. Note that the values in parentheses are differences between `CUDA` and `CUDA` with fixed augmentation order (Random order − Fixed order).

| | Category | CE | CE-DRW | LDAM-DRW | BS | RIDE |
|---|---|---|---|---|---|---|
| | Many | 71.6 | 64.3 | 67.3 | 63.3 | 69.2 |
| Random order (Ours) | Med | 42.3 | 49.2 | 50.4 | 48.4 | 52.8 |
| | Few | 9.4 | 26.7 | 21.4 | 28.7 | 27.3 |
| | All | 42.7 | 47.7 | 47.6 | 47.7 | 50.7 |
| | Many | 70.5 (-1.1) | 62.8 (-1.5) | 66.9 (-0.4) | 62.5 (-0.8) | 68.2 (-1.0) |
| Fixed order | Med | 43.0 (+0.7) | 50.2 (+1.0) | 49.7 (-0.7) | 48.3 (-0.1) | 53.5 (+0.7) |
| | Few | 9.0 (-0.4) | 27 (+0.3) | 21.9 (+0.5) | 29.6 (+0.9) | 26.9 (-0.4) |
| | All | 42.4 (-0.3) | 47.7 (+0.0) | 47.4 (-0.2) | 47.7 (+0.0) | 50.5 (-0.2) |

Table 11: The performance comparison between train network with randomly selected five and ten augmentation operations for every iteration and our proposed `CUDA`.

| | Category | CE | CE-DRW | LDAM-DRW | BS | RIDE |
|---|---|---|---|---|---|---|
| | Many | 66.2 | 62.8 | 62.8 | 61.6 | 67.7 |
| Vanilla | Med | 37.3 | 41.7 | 42.3 | 42.3 | 51.5 |
| | Few | 8.2 | 16.2 | 19 | 23 | 26.7 |
| | All | 38.7 | 41.4 | 42.5 | 43.3 | 49.7 |
| | Many | 68.9 | 56.9 | 64.2 | 59.5 | 64 |
| Randomly selected 5 augmentations | Med | 35 | 48.2 | 42.7 | 48.2 | 44.8 |
| | Few | 3.7 | 25.6 | 16.3 | 23 | 20 |
| | All | 37.5 | 44.5 | 42.3 | 44.6 | 44.1 |
| | Many | 61.7 | 51.2 | 57.9 | 54 | 57.7 |
| Randomly selected 10 augmentations | Med | 25.7 | 43.2 | 34.7 | 41.2 | 38.3 |
| | Few | 1.3 | 20.6 | 12.7 | 16.8 | 16.6 |
| | All | 31.0 | 39.2 | 36.2 | 38.4 | 38.6 |
| | Many | 71.6 | 64.3 | 67.3 | 63.3 | 69.2 |
| `CUDA` | Med | 42.3 | 49.2 | 50.4 | 48.4 | 52.8 |
| | Few | 9.4 | 26.7 | 21.4 | 28.7 | 27.3 |
| | All | 42.7 | 47.7 | 47.6 | 47.7 | 50.7 |

# E EXPERIMENTAL SETTING OF FIGURE 5D

To further analyze the impact of curriculum, we compare `CUDA` with the performance of previous hyper-parameter search algorithms and auto-augmentation methods, especially DADA (Li et al., 2020b). We describe each setting in detail as follows.

**Baseline.** This is the case of training with standard data augmentation that consists of random cropping and probabilistic horizontal flip.

**Hyper-parameter search.** We utilize the strength score-based augmentation module in `CUDA` to verify the hyper-parameter search. In other words, samples in each class utilize $K$ augmentation operations. Therefore, we search the class-wise augmentation on the search space $K^N$ where $N$ is the number of classes. We leverage the hyper-parameter searching open-source library, Ray (Liaw et al., 2018), for search $K^N$ space efficiently. Among various search modules, we utilize the Hyper-OptSearch module, which is the implementation of the Tree-structured Parzen Estimator (Bergstra et al., 2013). Moreover, for fast search, we use the Asynchronous Successive Halving Algorithm (ASHA) (Li et al., 2020a). We run $1,000$ trials for each algorithms which spends almost 20 GPU hours (*i.e.,* $\times 80$ overhead compare to CUDA).

**Researched DADA operation on imbalanced CIFAR.** Because the officially offered policies on CIFAR by Li et al. (2020b) are searched for a balanced CIFAR dataset, we have to re-search the augmentation policy for the imbalanced dataset. We utilize the official code of DADA and replace the dataloader to re-search the operations. It spends $48$ minutes for searching the augmentation policy ($\times 8.6$ than the overhead of CUDA). Despite this additional overhead, DADA outputs worse performance than `CUDA` (even `CUDA` without curriculum case). This is because (1) DADA does not consider class-wise augmentation and (2) it does not consider the impact of class imbalance.

**CUDA without curriculum** To verify the impact of curriculum itself, we ran the following steps. (1) We conduct experiments with CUDA and get the strength of data augmentation for each class at the final epoch. (2) We re-train the network from scratch by using the strength parameter obtained from (1).

# F    FURTHER ANALYSES

To get better understanding, we conduct several analyses for our proposed method, CUDA.

## F.1    FURTHER ANALYSIS ON LoL SCORE

In this section, we conduct experimental ablation studies to understand the performance gain of our proposed method, CUDA.

**Suitability of LoL score as metric for class-wise difficulty.** The superiority of LoL score is to measure the difficulty metric based on the augmentation strength for each class, which is motivated by our main findings. To verify the suitability of LoL score as a metric for class-wise difficulty, we compared CUDA and the case where LoL score is replaced by the score in Sinha et al. (2022). As same with our proposed method, we increase the strength parameter when the score in Sinha et al. (2022) is larger than the same threshold $\gamma = 0.6$. Table 12 summarizes the results that our LoL score showed performance improvement compared to the case of Sinha et al. (2022). From the results, we can conclude that this improvement comes from the characteristic of LoL score that is directly related to augmentation strength.

Table 12: The performance comparison between the scores for determining strength parameter, Sinha et al. (2022) and LoL score (ours).

|  | Category | CE | CE-DRW | LDAM-DRW | BS | RIDE |
|---|---|---|---|---|---|---|
| Sinha et al. (2022) | Many | 68.4 | 59.7 | 62.0 | 59.7 | 67 |
|  | Med | 42.5 | 48.8 | 48.7 | 47.0 | 52.1 |
|  | Few | 11.6 | 27.3 | 25.4 | 32.0 | 26.7 |
|  | All | 42.3 | 46.1 | 46.4 | 46.9 | 49.6 |
| LoL score | Many | 71.6 | 64.3 | 67.3 | 63.3 | 69.2 |
|  | Med | 42.3 | 49.2 | 50.4 | 48.4 | 52.8 |
|  | Few | 9.4 | 26.7 | 21.4 | 28.7 | 27.3 |
|  | All | 42.7 | 47.7 | 47.6 | 47.7 | 50.7 |

**Effect of random sampling for computing LoL score** To implement the computation of LoL score efficiently, we randomly selected the instances for each class. The reason for using random sampling to compute $V_{\text{Correct}}$ is that we want to measure how much the model learns entire information for each class. To understand the effect of random sampling, we compare our random sampling method to sampling instances with larger (or smaller) losses. Table 13 describes the comparison of performance between various sampling strategies. As shown in the results, if CUDA measures the degree of learning with only easy samples (the samples with small losses), CUDA increases the strength of augmentation too quickly and generates performance degradation. Therefore, it is a better way to grasp the degree of learning for each class without prejudice through uniform random sampling. Furthermore, computing loss for all samples for sorting them at the beginning of each epoch requires $\times 1.5$ times of computation overhead than our method.

**Numerical values of LoL score dynamics.** We provide the numerical values for Figure 4 that is, the average values (for every 20 epochs) of LoL score for the classes with indices 1-10 and the classes with indices 91-100. From the numerical values, we can easily understand the explanation which is discussed in Section 4.

Table 13: The performance comparison between the large loss sample selection, small loss sample selection, and random selection (ours).

|  | Category | CE | CE-DRW | LDAM-DRW | BS | RIDE |
|---|---|---|---|---|---|---|
| Larger Loss | Many | 67.0 | 61.6 | 63.1 | 59.9 | 67.9 |
|  | Med | 37.1 | 45.2 | 45.7 | 42.4 | 51.2 |
|  | Few | 7.3 | 20.3 | 20.3 | 23.3 | 25.8 |
|  | All | 38.6 | 43.5 | 44.2 | 42.8 | 49.4 |
| Smaller Loss | Many | 53.0 | 53.4 | 54.5 | 51.2 | 59.3 |
|  | Med | 24.7 | 33.0 | 33.9 | 36.1 | 38.4 |
|  | Few | 24.2 | 32.9 | 33.7 | 35.4 | 38.2 |
|  | All | 41.6 | 44 .0 | 45.5 | 45.7 | 49.8 |
| Random (Ours) | Many | 71.6 | 64.3 | 67.3 | 63.3 | 69.2 |
|  | Med | 42.3 | 49.2 | 50.4 | 48.4 | 52.8 |
|  | Few | 9.4 | 26.7 | 21.4 | 28.7 | 27.3 |
|  | All | 42.7 | 47.7 | 47.6 | 47.7 | 50.7 |

Table 14: The averaged LoL score of top 10 classes (class indices with 1-10) and bottom classes (class indices with 91-100) for every 20 epochs.

|  | Class / Epoch | 20 | 40 | 60 | 80 | 100 | 120 | 140 | 160 | 180 | 200 |
|---|---|---|---|---|---|---|---|---|---|---|---|
| CE | Top 10 | 0.5 | 0.8 | 1.7 | 2 | 2.6 | 2.7 | 3.2 | 3.2 | 3.4 | 3.7 |
|  | Bottom 10 | 0.0 | 0.0 | 0.0 | 0.0 | 0.0 | 0.1 | 0.0 | 0.1 | 0.0 | 0.1 |
| CE-DRW | Top 10 | 0.8 | 1.3 | 2 | 2.5 | 1.6 | 2.6 | 1.8 | 2.6 | 2.5 | 2.6 |
|  | Bottom 10 | 0.0 | 0.0 | 0.0 | 0.0 | 0.0 | 0.1 | 0.0 | 0.0 | 1.2 | 1.4 |
| LDAM-DRW | Top 10 | 1.8 | 3.4 | 3.3 | 3 | 3.4 | 3.2 | 3.1 | 3.6 | 4.1 | 4.3 |
|  | Bottom 10 | 0.0 | 0.0 | 0.1 | 0.0 | 0.0 | 0.1 | 0.0 | 0.0 | 1.8 | 1.4 |
| BS | Top 10 | 1.1 | 1.1 | 1.6 | 2.4 | 2.1 | 2.5 | 2.5 | 2.8 | 3.9 | 4.0 |
|  | Bottom 10 | 0.1 | 0.1 | 0.0 | 0.1 | 0.6 | 1.1 | 1.0 | 0.7 | 1.7 | 1.4 |
| RIDE | Top 10 | 0.8 | 1.2 | 1.6 | 1.7 | 2.1 | 1.8 | 1.3 | 1.1 | 2.5 | 2.6 |
|  | Bottom 10 | 0.0 | 0.0 | 0.0 | 0.0 | 0.0 | 0.0 | 0.0 | 0.0 | 0.7 | 0.8 |

## F.2 ANALYSIS THE CASE OF WITHOUT CLASS-WISE

To examine the validity of class-wise augmentation of CUDA, we apply the CUDA with the same strength of DA for all classes. Instead of computing LoL score class-wisely, we computed only one LoL score for the entire dataset by uniformly random sampling instances in the training dataset regardless of class. Table 15 shows the significant performance degradation of CUDA without class-wise augmentation compared to CUDA. This is because, without class-wise augmentation, we cannot allocate the appropriate strength of augmentation to each class.

Table 15: The performance comparison between different augmentation strategy on CIFAR-100-LT with imbalance ratio 100. Note that values in parentheses are differences of CUDA w/o class-wise with vanilla (vanilla - CUDA w/o class-wise) or CUDA (CUDA- CUDA w/o class-wise).

|  | Category | CE | CE-DRW | LDAM-DRW | BS | RIDE |
|---|---|---|---|---|---|---|
| CUDA w/o class-wise | Many | 69.0 | 62.7 | 65.2 | 62.2 | 67.7 |
|  | Med | 38.7 | 47.3 | 47.6 | 44.7 | 52.2 |
|  | Few | 6.7 | 23.5 | 20.7 | 25.5 | 26.4 |
|  | All | 39.7 | 45.5 | 45.7 | 44.9 | 49.9 |
| Vanilla | Many | 66.2 (-2.8) | 62.8 (+0.1) | 62.8 (-2.4) | 61.6 (-0.6) | 67.7 (+0.0) |
|  | Med | 37.3 (-1.4) | 41.7 (-5.6) | 42.3 (-5.3) | 42.3 (-2.4) | 51.5 (-0.7) |
|  | Few | 8.2 (+1.5) | 16.2 (-7.3) | 19.0 (-1.7) | 23.0 (-2.5) | 26.7 (+0.3) |
|  | All | 38.7 (-1.0) | 41.4 (-4.1) | 42.5 (-3.2) | 43.3 (-1.6) | 49.7 (-0.2) |
| CUDA (Ours) | Many | 71.6 (+2.6) | 64.3 (+1.6) | 67.3 (+2.1) | 63.3 (+1.1) | 69.2 (+1.5) |
|  | Med | 42.3 (+3.6) | 49.2 (+1.9) | 50.4 (+2.8) | 48.4 (+3.7) | 52.8 (+0.6) |
|  | Few | 9.4 (+2.7) | 26.7 (+3.2) | 21.4 (+0.7) | 28.7 (+3.2) | 27.3 (+0.9) |
|  | All | 42.7 (+3.0) | 47.7 (+2.2) | 47.6 (+1.9) | 47.7 (+2.8) | 50.7 (+0.8) |

