# OpenReview forum: "CUDA: Curriculum of Data Augmentation for Long-tailed Recognition"
_ICLR.cc/2023/Conference — ICLR 2023 notable top 25%_

### Official Review · Reviewer_QwML · 2022-10-23

**Confidence:** 5
**Correctness:** 3
**Technical Novelty And Significance:** 3
**Empirical Novelty And Significance:** 3
**Recommendation:** 6

**Clarity, Quality, Novelty And Reproducibility:**

- The paper is mostly clear except the points I listed above.
- The quality of the paper in terms of its writing, the technical soundness, and the support provided by the experiments is good.
- Even though the idea of using different strength of data augmentation itself and the curriculum learning itself are well-known, I think the design presented in this paper to combine them to better handle long-tailed problems has good novelty.
- Since the code is provided, I believe the results are reproducible though I have not tried to do so by myself.

**Strength And Weaknesses:**

## Strength
1. The idea of changing the strength of data augmentation for each class during the training depending on their performance is reasonable and interesting. The idea is so simple that it can be easily integrated with other methods for long-tailed learning as shown in the paper.
1. The proposed method is evaluated on popular datasets for long-tailed learning and shows consistently good performance.
1. The analysis provided in section 4.3 helps readers understand the characteristics of the proposed method.
1. The paper is mostly well written and easy to follow.

## Weakness
1. In my view, the important contributions of the paper are two-fold: (1) the paper presented a new way to estimate the performance of each class during training, and (2) the paper presented the idea of changing the strength of augmentation according to the performance of each class during training. I think the paper becomes stronger if it could provide more in-depth analysis on which part of the two are essential.
    1. For (1) [A1] proposed to use validation data to estimate the difficulty of each class. Possibly the authors can use this method in the LoL process instead of the proposed one. This clarifies if it is the proposed LoL process that makes the overall performance better or an alternative method can also be used in the LoL process.
    1. For (2), even though the authors compare the proposed method with other augmentation methods, it is not clear if the overall performance gain comes from the proposed curriculum, or the selected set of data augmentation and its strength presented in Appendix D. The last paragraph of Section 4 “Impact of curriculum” may be the one for this analysis, but I could not judge if the proposed curriculum is actually effective since there is no detailed explanation on other methods. For example, what is the “vanilla algorithm”? Which “hyperparameter optimization” method is used? Please elaborate “the final augmentation strength of CUDA” more. What happens if the kinds of augmentation and its strength is randomly sampled from the sets presented in Appendix D in each step? I am afraid that the performance gain presented in Table 1-3 actually come from just a richer set of data augmentation.
1. It is not clear what “the augmentation policies” in the 3rd line from the bottom of p8 means. Is it the augmentation policies of the existing methods or the proposed method? Even though the authors state that ”CUDA is computationally efficient” in the last line, I doubt that it is not necessarily the case because the proposed method needs to run multiple inferences to calculate $V_{Correct}$.

[A1] Sinha+, Class-Difficulty Based Methods for Long-Tailed Visual Recognition, IJCV 2022

Minor points.
1. I needed to read several times and make some guesses to understand what each graph in Figure 1 shows. I think the 6 graphs shown in the top row represent the accuracies of majority classes (class index 0-49) while the 6 graphs in the bottom row show those of minority classes (class index 50-99). I think it is better to add clearer and more detailed explanation in the caption or the figure itself.
1. Figure 3: It is necessary to clearly indicate in the caption that the bottom row shows feature alignment gain, not feature alignment.
1. It is good that the dynamics of the LoL score is shown in Figure 4, and it shows reasonable results. I wonder if it has actually correlate with the accuracy. The paper may become stronger if the analysis on the correlation between the estimated LoL score and validation or test accuracy is added.



**Summary Of The Paper:**

This paper proposes a new data-augmentation strategy based on curriculum learning for long-tail problems. The key idea is to estimate the appropriate strength of data augmentation needed for each class during training. The proposed method was evaluated on widely used datasets and achieved favorable performance.

**Summary Of The Review:**

I think the key idea of the paper is reasonable and the empirical evidence provided in the experiments are sufficient. Although the paper has some room for improvement, I think the paper has already enough contribution to the community.

---

> ### Author Response · Authors · 2022-11-11
> **Response to Reviewer QwML (3/3)**
>
>
> ---
> **Q6) Is there a correlation between LoL score and performance?**
>
> A6) To answer the question, we compare the CE-DRW and BS, which have similar, e.g., 47.7\%,  end performance (CE-DRW + CUDA and BS + CUDA) but different LoL score dynamics as shown in Figure 4. We describe the correlation between LoL score and performance for each method as follows:
> CE-DRW: We can see that the majority classes increase their LoL score fastly and largely, while minority classes do not increase their score until conducting DRW. However, in DRW phase (after the 160th epoch), the LoL scores of minority classes increases. We can conclude that, compared to BS, higher LoL scores of majority classes significantly improve the few category performances (+10.6% on average).
> BS: Unlike CE-DRW, the LoL score of all classes increases similarly due to the logit adjustment of BS. Compared to CE-DRW, the amount of increment of LoL score for majority (or minority) is smaller (larger), respectively. Therefore, the performance improvement for the “few” category (+5.7%) is smaller than that of CE-DRW (+10.6%). Oppositely, the performance improvement of the “many” category (+1.8%) is larger than that of CE-DRW (+1.5%).
>
> From our observation, we can conclude that the LoL score of one category has a high correlation with the performance of the opposite classes. This is consistent with our main finding in Figure 1.
>
> ---
> **Q7) Minor typos**
>
> A7) We fixed the typos about feature alignment in Figure 3 in our revised manuscript. Moreover, we will focus on improving the quality of grammar through a professional editing service and enhance the clarification of our contribution.
>
> ---
> [A] Class-Difficulty Based Methods for Long-Tailed Visual Recognition. IJCV. 2022

---

> > ### Comment · Reviewer_QwML · 2022-11-14
> > **My concerns addressed**
> >
> > I appreciate the feedback from the authors.
> > I have carefully read the authors' response and the other reviewers' comments.
> > I think my concerns are addressed and the paper became more convincing.
> > As such I vote for accepting this paper.

---

> > > ### Author Response · Authors · 2022-11-15
> > > **Thank you for your response**
> > >
> > > We appreciate your positive assessment of our work. Thanks to you, we were able to make our manuscript that could be delivered more clearly.

---

> ### Author Response · Authors · 2022-11-11
> **Response to Reviewer QwML (2/3)**
>
>
> ---
> **Q3) Concern about where superiority comes from**
>
> A3) To answer your concern that the performance gain came from just a richer set of data augmentation, we conducted the experiments with randomly sampled $K$ augmentation operations for every iteration as you suggested. The results are as follows:
>
>
> |           | category |  CE  | CE-DRW | LDAM-DRW |  BS  | RIDE |
> |-----------|:--------:|:----:|:------:|:--------:|:----:|:----:|
> | Vanilla   |   Many   | 66.2 |   62.8 |     62.8 | 61.6 | 67.7 |
> |           |    Med   | 37.3 |   41.7 |     42.3 | 42.3 | 51.5 |
> |           |    Few   |  8.2 |   16.2 |     19.0 |  23.0 | 26.7 |
> |           |    ALL   | 38.7 |   41.4 |     42.5 | 43.3 | 49.7 |
> | Random 5 $(K=5)$  |   Many   | 68.9 |   56.9 |     64.2 | 59.5 | 64.0 |
> |           |    Med   | 35.0 |   48.2 |     42.7 | 48.2 | 44.8 |
> |           |    Few   |  3.7 |   25.6 |     16.3 | 23.0 | 20.0 |
> |           |    ALL   | 37.5 |   44.5 |     42.3 | 44.6 | 44.1 |
> | Random 10 $(K=10)$ |   Many   | 61.7 |   51.2 |     57.9 |   54 | 57.7 |
> |           |    Med   | 25.7 |   43.2 |     34.7 | 41.2 | 38.3 |
> |           |    Few   |  1.3 |   20.6 |     12.7 | 16.8 | 16.6 |
> |           |    ALL   | 31.0 |   39.2 |     36.2 | 38.4 | 38.6 |
> | CUDA $(K=22)$      |   Many   | 71.6 |   64.3 |     67.3 | 63.3 | 69.2 |
> |           |    Med   | 42.3 |   49.2 |     50.4 | 48.4 | 52.8 |
> |           |    Few   |  9.4 |   26.7 |     21.4 | 28.7 | 27.3 |
> |           |    ALL   | 42.7 |   47.7 |     47.6 | 47.7 | 50.7 |
>
>
> Note that Random 5 and Random 10 denote the conduct of five and ten randomly sampled augmentations for every iteration. As shown in the above table, while Random 10 generates the most diversifying images, the network trained with this showed the worst performance even lower than vanilla (which only trained with standard data augmentations; random cropping, and horizontal flipping). Note that CUDA achieves the best performance among all cases.
>
> ---
> **Q4) Meaning of the “augmentation policy” on page 8 and the overhead due to the $V_{correct}$ operation**
> A4) First of all, the “augmentation policy” in page 8 was wrongly written, and it should be changed to “automated data augmentation methods” as you mentioned. We are sorry for the confusion caused by our mistake.
>
> When understood as a modified “automated data augmentation method”, the augmentation policy searching time of CUDA, i.e., overhead to compute $V_{correct}$, is significantly low compared to the previous automated augmentation searching cost. A comparison of computation time to find the augmentation policy is described in Appendix E. In the case of DADA, which is the fastest automated data augmentation method, it takes 48 min, which is 8.6 times longer than CUDA. Thus, we argue that CUDA is computationally efficient compared to the other automated augmentation methods.
>
> ---
> **Q5) Readability of Figure 1**
>
> A5) We rephrase Figure 1 and its caption to describe our finding much more clearly. Thank you for pointing out for improving the interpretability of Figure 1. Here are our revisions of the manuscript.
> We added additional indicators to Figure 1 to clarify the meaning of each heatmap and each point.
> We denote the metric of performance as averaged accuracy and describe the meaning of each heatmap and each point.
> We think that the revised Figure 1 and its caption can explain our finding. If there is anything we miss or wrongly understand your concerns, please give us additional comments, which will greatly help to improve the readability of many readers of this manuscript in the future.

---

> ### Author Response · Authors · 2022-11-11
> **Response to Reviewer QwML (1/3)**
>
> We are very grateful for your constructive comments. We express your concerns that we understood and answer them as follows. If there is any worry that we misunderstood or additional questions, we look forward to being able to answer them with further comments from you.
>
> ---
> **Q1) Suitability of LoL score as metric for class-wise difficulty**
>
> A1) The superiority of LoL score is to that it measures the difficulty metric based on the augmentation strength, which is our main scope. Therefore, it can provide appropriate augmentation strength for each class. We compare the performance between the proposed CUDA with our LoL score and the case where the LoL score is replaced by the score in [A], suggested by the reviewer. In the case of [A], we increase the strength parameter when the score in [A] is larger than the same threshold $\gamma = 0.6$.
>
>
> |                  | category |  CE  | CE-DRW | LDAM-DRW |  BS  | RIDE |
> |------------------|:--------:|:----:|:------:|:--------:|:----:|:----:|
> | Score in [A]     |   Many   | 68.4 |   59.7 |     62.0 | 59.7 | 67.0 |
> |                  |    Med   | 42.5 |   48.8 |     48.7 | 47.0 | 52.1 |
> |                  |    Few   | 11.6 |   27.3 |     25.4 | 32.0 | 26.7 |
> |                  |    ALL   | 42.3 |   46.1 |     46.4 | 46.9 | 49.6 |
> | LoL score (ours) |   Many   | 71.6 |   64.3 |     67.3 | 63.3 | 69.2 |
> |                  |    Med   | 42.3 |   49.2 |     50.4 | 48.4 | 52.8 |
> |                  |    Few   |  9.4 |   26.7 |     21.4 | 28.7 | 27.3 |
> |                  |    ALL   | 42.7 |   47.7 |     47.6 | 47.7 | 50.7 |
>
>
> As shown in the above, our LoL score shows improved performance compared to the case of [A], and we think that this improvement comes from the characteristic of LoL score, which is directly related to augmentation strength.
>
> ---
> **Q2) Detail information of Figure 5(d)**
>
> A2) First of all, we already wrote the experimental settings of Figure 5(d) in Appendix E but forgot to mention it in the main manuscript. Here, we additionally introduce the explanation of Appendix E to address the concerns of the reviewer a little more clearly as follows:
> - Vanilla (changed to “Baseline" in the revised manuscript): This is the case of baseline, i.e., training with standard data augmentation, which consists of random cropping and probabilistic horizontal flip. To clarify, we modify the legend in Figure 5(d) and add corresponding information in Appendix E.
> - HyperOpt: We can think of the augmentation strength for each class as a hyperparameter; thus, we tried to tune them via hyperparameter tuning algorithm. We used the open-source library, Ray, with the HyperOptSearch function. This module utilizes the Tree-structured Parzen Estimator method. In addition, Asynchronous Successive Halving Algorithm (ASHA scheduler) was used to speed up the training. We ran 1,000 trials for searching with CE loss function, and it spend about 20 GPU hours (x80 overhead more than CUDA).
> - DADA for CIFAR-100-LT: Because the officially released policy of DADA is searched on the original balanced CIFAR-100, we ran the DADA code on the CIFAR-100-LT dataset to obtain a proper augmentation policy for the imbalanced case. We used the official code of DADA and set the hyperparameter values as the same with DADA paper for the balanced case. It takes 48 minutes, which is 8.6 times longer than CUDA overhead.
> - CUDA without curriculum: To verify the impact of curriculum itself, we ran the following steps. (1) We conduct experiments with CUDA and get the strength of data augmentation for each class at the final epoch. (2) We re-train the network from scratch by using the strength parameter obtained from (1).

---

### Official Review · Reviewer_nPVy · 2022-10-23

**Confidence:** 5
**Correctness:** 3
**Technical Novelty And Significance:** 3
**Empirical Novelty And Significance:** 3
**Recommendation:** 6

**Clarity, Quality, Novelty And Reproducibility:**

The paper is mostly clear and easy to understand with some parts in method and experiment for more clarity.

The novelty is sort of limited due to that dataset-specific data augmentation exists.

The reproducibility seems good since the method design is not complex.

The overall quality is limited at the current form, considering the weaknesses as listed above.





**Strength And Weaknesses:**

**Strengths**

- Data augmentation is an interesting dimension to study class imbalance in general.

- The proposed method is simple and reasonably effective in the sense that it exploits the empirical analysis.


**Weaknesses**

- The contribution of "the first to suggest a class-wise augmentation method to find a proper augmentation strength for class imbalance problem" is over claimed. As this is a special case of exiting dataset-specific data augmentation methods such as  (Cheung & Yeung, ICLR 2022).

- In Introduction, the findings on the data augmentation and class performances are given without proper and concise explanation. This is fine only when they are intuitive and there is no need for explaining, but it seems not.

- Figure 1 needs more interpretation in the caption. e.g. What is the metric for the grid charts

- Method:
> 1)  A few hyper-parameters are introduced, and how to tune them is not clear.
> 2) What means by curriculum of DA?

- Experiments:
> 1) Given that this paper is about data augmentation, it is useful and necessary to compare with existing dataset-specific data augmentation search methods such as (Cheung & Yeung, ICLR 2022). This is missing now.
> 2) The performance gain on larger datasets (ImageNet, iNaturelist 2018) in Table 2 is smaller. This seems that class-wise data augmentation will get less useful along the scale dimension. This is a key issue for experimental evaluation.
> 3) Fig 5 (d): This shows that the most performance margin is from the use of curriculum. Given this, what is the performance for the case of w/o CUDA & w. Curriculum?

**Summary Of The Paper:**

This paper deals with class imbalance from the data augmentation perspective. It is based on and motivated by an analysis on the relationship between data augmentation degree and performance per class.

**Summary Of The Review:**

A OKish work on class imbalanced learning with some limited evaluations and also likely limited scalability issue.

Some clarity issues need to be addressed for the final judgement.

---

> ### Author Response · Authors · 2022-11-11
> **Response to Reviewer nPVy (5/5)**
>
>
> ---
> [A] Curriculum Learning. ICML. 2009
>
> [B] When Do Curricula Work?. ICLR. 2021
>
> [C] Curriculum Learning by Dynamic Instance Hardness. NeurIPS. 2020
>
> [D] AdaAug: Learning Class- and Instance-adaptive Data Augmentation Policies. ICLR. 2022
>
> [E] Long-tailed Recognition by Routing Diverse Distribution-Aware Experts. ICLR. 2021
>
> [F] The Majority Can Help the Minority: Context-rich Minority Oversampling for Long-tailed Classification. CVPR. 2022
>
> [G] Balanced Contrastive Learning for Long-Tailed Visual Recognition. CVPR. 2022

---

> > ### Comment · Reviewer_nPVy · 2022-11-19
> > **Post-rebuttal review**
> >
> > Thanks for the detailed response and extra experiments/discussion by the authors.
> >
> > The feedback has well addressed my concerns and issues in particular on parameter tuning, performance gain across scales, clarity improvement.
> >
> > Along with addressing the comments from the other reviewers, this paper has been clearly improved in many aspects. Overall, the idea of class-wise data augmentation in a curriculum during training is an interesting dimension for tackling class imbalance, and this is one of the most dedicated study.

---

> > > ### Author Response · Authors · 2022-11-21
> > > **Thank you for your response**
> > >
> > > Dear Reviewer nPVy,
> > >
> > > We sincerely appreciate you taking the time to read through our response.
> > >
> > > We were able to create much clearer manuscript thanks to your constructive and insightful feedbacks.
> > >
> > > Sincerely,
> > >
> > > Authors

---

> ### Author Response · Authors · 2022-11-11
> **Response to Reviewer nPVy (4/5)**
>
>
> ---
> **Q6) Missing proper and concise analysis of the findings in the introduction section**
>
> A6) For a clearer explanation for our counter-intuitive finding, we slightly modified the sentence that described the concise intuition of the phenomenon as follows:
>
> *“To explain this finding, we further find that strongly augmented classes get diversified feature representation, preventing the growth of the norm of a linear classifier for corresponding classes. As a result, the softmax outputs of the strongly augmented classes are reduced, and thus the accuracy of those classes decreases.”*
>
> As mentioned in the original manuscript, you can find detailed explanations with experimental evidence in Appnedix A.
>
> ---
> **Q7) How did we tune the hyper-parameters of CUDA?**
>
> A7) We did not tune the hyper-parameters extensively. However, we give a guideline to select the hyper-parameters as follows.
> - We set the hyperparameter $p_{\text{aug}} = 0.5$, $T= 10$, and $\gamma = 0.6$, respectively.
> - For $T$ (number of samples for updating LoL), we can set it according to the given computing resources, i.e., the largest T under computing resource constraint. This is because the performance improves as T increases from obtaining a definite LoL score by testing many samples.
> Our strategy for tuning $\gamma$ is to select the largest $\gamma$ in which at least one of LoL scores among all classes increases within 20 epochs. Here is the detailed tuning strategy of $\gamma$.
> - We initially set $\gamma$ as 0.6. However, for large-scale datasets, they fail to infer even the easier-to-learn majority classes because they are difficult to learn due to the huge size of the training dataset. Therefore, we decrease the threshold $\gamma$ by 0.1 points whenever it fails to raise any of LoL scores for the first 20 training epochs. We conduct this search on CE with CIFAR-100-LT with Imbalance ratio (IR) 100 and using the same $\gamma$ value for the other algorithms with remaining IR settings. Also, we conduct this search rule on ImageNet-LT with CE and use the same value to the other large-scale dataset (iNaturalist 2018) with remaining algorithms.
> - For $p_{\text{aug}}$ (augmentation probability), we did not tune this hyperparameter. However, we offer the guideline how to tune the augmentation probability $p_{\text{aug}}$. As shown in Figure 5(a), the shape of $p_{\text{aug}}$ v.s. performance is concave. Thanks to concavity, we think that it is easy to find the optimal value for this hyper-parameter.
> Note that the reason for the concavity is because the decision of $p_{\text{aug}}$ value has a trade-off between preserving the information of the original image and exploring diversified images.
>
> In addition, as the cases of $\gamma=0.5$, $T=100$ or $p_{\text{aug}} = 0.6$ in Figure 5 (a), (b) and (c) show better performance than the 50.7 reported in Table 1, we believe that better performance can be expected if CUDA is tuned in more depth with proper parameter tuning algorithms. Furthermore, there could be a much better strategy to find the best hyper-parameter; however, this is beyond the scope of this work.
>
> ---
> **Q8) Concerns about low performance gain in large-scale benchmark (Table 2)**
>
> A8) We think that our improvement is comparable, by seeing the performance of the previous papers, such as RIDE [E], CMO [F], and BCL [G]. We summarize the performance gain of each algorithm in our implementation as follows:
>
> |             | CMO (CVPR’22)                          | BCL (CVPR’22)                     | CUDA (Ours)                            |
> |-------------|----------------------------------------|-----------------------------------|----------------------------------------|
> | ImageNet-LT | (RIDE -> CMO+RIDE)53.6 -> 54.0 (+0.4)  | (RIDE -> BCL)53.6 -> 55.6 (+2.0)  | (BCL-> CUDA+BCL) 55.6 -> 56.3 (+0.7)   |
> | iNaturalist | (RIDE -> CMO+RIDE) 71.6 -> 70.9 (-0.7) | (RIDE -> BCL) 71.6 -> 71.8 (+0.2) | (BCL -> CUDA+RIDE) 71.8 -> 72.4 (+0.6) |
>
>
> As shown in the above table, the gain of CUDA on ImageNet-LT (from 55.6 to 56.3 - i.e., +0.7) and iNaturalist (from 71.8 to 72.4 - i.e., +0.6) is comparable to the gain of previous works. For example, CMO and BCL obtain 0.4% and 1.4% on ImageNet-LT, respectively. Furthermore, in the iNaturalist case, CMO shows -0.7% degradation, and BCL obtains 0.6% improvement compared to RIDE on our implementation. Note that the official code of CMO did not contain CMO+RIDE case, thus we re-implement it by utilizing the case of CMO+* (e.g., CMO+BS). Therefore, we believe that the gain of CUDA is meaningful.

---

> ### Author Response · Authors · 2022-11-11
> **Response to Reviewer nPVy (3/5)**
>
>
> ---
> **Q4) Comparison with AdaAug [D], an augmentation method using an augmentation policy network**
>
> A4) We were aware of AdaAug [D] before submission (we already cited this paper in the original manuscript), and we re-implemented this method using the official code to compare to CUDA. The results we got are as follows.
>
>
> |                                                 | CE   | BS   |
> |-------------------------------------------------|------|------|
> | Vanilla                                         | 38.8 | 43.3 |
> | AdaAug (Aug. Net trained on reduced CIFAR-100)  | 40.4 | 45.9 |
> | AdaAug (Aug. Net trained on CIFAR100-LT IR 100) | 40.4 | 46.1 |
> | CUDA                                            | 42.7 | 47.7 |
>
> As shown in the above table, BS + CUDA outperformed BS + AdaAug by 1.6%.
> Note that our implementation was not wrong, by checking the performance of without augmentation cases which are very similar to the our reported values in the manuscript.
>
> However, the reason we did not report the results is that there is a reproducibility issue in the case of the original CIFAR-100 (achieved 77.7% which is worse than the reported value 80.2%), and we were worried about reporting the performance of AdaAug. Similar to our case, we found that there exists a reproducibility issue in the official github website that is raised by other researchers (achieved 90.86%, which is worse than reported value of 91.8% on SVHN dataset).
>
> ---
> **Q5) Lack of some information and interpretation in the caption of Figure 1**
>
> A5) We have rephrased Figure 1 and its caption to describe our finding much clearer. Thank you for pointing out the need for improving the interpretability of Figure 1. We addressed this as follows:
> We have added additional indicators to Figure 1 to clarify the meaning of each heatmap and each point.
> We have noted that the metric of performance is averaged accuracy and describe the meaning of each heatmap and each point in the caption of Figure 1.
> We think that the revised Figure 1 and its caption can explain our finding. If there is anything we miss or wrongly understand your concerns, please give us additional comments, which will greatly help to improve the readability of the manuscript in the future.

---

> ### Author Response · Authors · 2022-11-11
> **Response to Reviewer nPVy (2/5)**
>
>
> ---
> **Q2) What is the performance for the case of w/o CUDA & w. curriculum in Figure 5 (d)?**
>
> A2) As we mentioned the meaning of “curriculum of DA” in Q1, it is infeasible to use the curriculum without using CUDA, which the reviewer asked for. If our understanding of the question is correct, the case of (without CUDA & with curriculum) seems to be interpreted as the following two cases:
>
> - Conventional curriculum learning: Our method has a low correlation with conventional curriculum learning methods [A-C], which adjust batch configuration based on the per-sample difficulty. Unlike them, CUDA provides adaptively proper augmentation for each class based on class-wise difficulty, without a curriculum of batch configuration.
>
> - CUDA without class-wise augmentation: If there is no class-wise part of CUDA, we cannot implement our intuition that the majority class has to be more difficult and the minority class has to be easier via strength of data augmentation. Therefore, we can predict performance degradation. The results of this experiment are shown in the table below.
>
> |                     | category |      CE     |    CE-DRW   |   LDAM-DRW  |      BS     |     RIDE    |
> |---------------------|:--------:|:-----------:|:-----------:|:-----------:|:-----------:|:-----------:|
> | CUDA w/o class-wise | Many     |        69.0 |        62.7 |        65.2 |        62.2 |        67.7 |
> |                     | Med      |        38.7 |        47.3 |        47.6 |        44.7 |        52.2 |
> |                     | Few      |         6.7 |        23.5 |        20.7 |        25.5 |        26.4 |
> |                     | All      |        39.7 |        45.5 |        45.7 |        44.9 |        49.9 |
> | Vanilla             | Many     | 66.2 (-2.8) | 62.8 (+0.1) | 62.8 (-2.4) | 61.6 (-0.6) | 67.7 (+0.0) |
> |                     | Med      | 37.3 (-1.4) | 41.7 (-5.6) | 42.3 (-5.3) | 42.3 (-2.4) | 51.5 (-0.7) |
> |                     | Few      |  8.2 (+1.5) | 16.2 (-7.3) | 19.0 (-1.7) | 23.0 (-2.5) | 26.7 (+0.3) |
> |                     | All      | 38.7 (-1.0) | 41.4 (-4.1) | 42.5 (-3.2) | 43.3 (-1.6) | 49.7 (-0.2) |
> | CUDA (Ours)         | Many     | 71.6 (+2.6) | 64.3 (+1.6) | 67.3 (+2.1) | 63.3 (+1.1) | 69.2 (+1.5) |
> |                     | Med      | 42.3 (+3.6) | 49.2 (+1.9) | 50.4 (+2.8) | 48.4 (+3.7) | 52.8 (+0.6) |
> |                     | Few      |  9.4 (+2.7) | 26.7 (+3.2) | 21.4 (+0.7) | 28.7 (+3.2) | 27.3 (+0.9) |
> |                     | All      | 42.7 (+3.0) | 47.7 (+2.2) | 47.6 (+1.9) | 47.7 (+2.8) | 50.7 (+0.8) |
>
> Note that values in parentheses are differences of CUDA w/o class-wise with vanilla (vanilla - CUDA w/o class-wise) or CUDA (CUDA - CUDA w/o class-wise). As shown in the above table, the performance of the “few” category is significantly reduced: in particular, the average validation accuracy is also degraded compared to the class-wise approach.
>
> ---
> **Q3) Is our claim that “CUDA is the first class-wise augmentation method to find a proper augmentation strength for class imbalance problem”  over-claim?**
>
> A3) To the best of our knowledge, there have been studies looking for class-wise appropriate augmentation in the BALANCED dataset, but no effort has been made to solve ***the class imbalance problem by adjusting the strength of the augmentation***. In the experimental settings in the AdaAug [D], mentioned by the reviewer, the goal and evaluation of this work are basically to focus on balanced datasets. To make the statement a little clearer, we rephrase the statement to “CUDA is the first trial to find a proper augmentation strength ***for each class under the class imbalance problem***.”

---

> ### Author Response · Authors · 2022-11-11
> **Response to Reviewer nPVy (1/5)**
>
> We are very grateful for your constructive comments. We express your concerns that we understood and answer them as follows. If there is any worry that we misunderstood or additional questions, we look forward to being able to answer them with further comments from you. We answer the questions by reordering them in the order in which we thought they were highly correlated to help your understanding.
>
> ---
> **Q1) What is the meaning of the term “curriculum of DA (CUDA)”?**
>
> A1) The term “curriculum of DA” means “the learning difficulty is adjusted through the strength of data augmentation to provide a curriculum that fits the learned level of the training model.” The word “curriculum” was used in [A-C] to mean “a meaningful sequence that shows gradually more complex ones.” and we borrowed it. Our CUDA provides class-wise adaptive augmentation by combining (1) DA with a strength parameter, (2) level-of-learning (LoL). “CUDA” is a newly defined terminology in our paper, as mentioned in page 2 (the contribution paragraph of Sec 1) and page 5 (the Curriculum of DA paragraph of Sec 3.2).

---

### Official Review · Reviewer_URnx · 2022-10-25

**Confidence:** 3
**Correctness:** 4
**Technical Novelty And Significance:** 3
**Empirical Novelty And Significance:** 3
**Recommendation:** 8

**Clarity, Quality, Novelty And Reproducibility:**

The paper is clear and easy to follow along with a good motivation behind the approach. The work seems novel to me.

**Strength And Weaknesses:**

Strengths -
1. The paper is very well written and easy to flow along.
2. The motivation in the paper about how doing augmentations for some classes affects the performance of non-augmented classes is really useful.
3. Extensive experiments are conducted on top of existing baselines and over different datasets and the gains achieved show the usefulness of the approach.
4. I really like the comparison against fixed augmentations such as AutoAugment and RandAugment. Such comparisons show the efficacy of the proposed class-dependent augmentation strategy.
5. The comparison against doing a curriculum search using hyperparameter optimization algorithms is also neat and useful.

Weakness -
1. Since for computing the LoL the paper uses only a subset of the data from that class, I am curious to see the variance in the performance. However, none of the tables report the variance in the performance but only the mean result across 3 trials.

2. The paragraph "Dynamics of LoL score" needs more emphasis. Specifically, it would be interesting to see some statistics of what are the LoL scores for the majority classes and the minority classes after the end of the training. While figure 4 has the plots, it is difficult to conclude something from there. Can authors include some statistics, such as the mean LoL score for the top k majority and minority classes respectively?

3. How do the authors select which subset of data to use for LoL computation? Is it completely random? Can the authors also report what happens when the samples with the highest/lowest training loss and instead chosen?


**Summary Of The Paper:**

The paper proposes to use a class-dependent augmentation strategy for tackling the class imbalance problem. The paper initially observes that by using a class-dependent augmentation where only the majority classes are augmented, the performance of the minority classes improves but that over the majority class regresses. Thus, the paper proposes to learn a curriculum for class-dependent augmentation strength, with the idea being that if the model is performing good over a certain class for an augmentation strength then the strength can be increases and reduced otherwise. They propose level-of-learning, which each class computes the augmentation strength to be used for that epoch. At each epoch, it is either incremented or decremented based on the model's performance over a subset of data for that class over all the augmentation strengths up to the current one.

Empirically the proposed method achieves a good boost on top of several different existing long-tail tackling approaches over different datasets. To show that their approach leads to a good balance between the majority and minorty classes, the authors plot the variance of weight L1-norm of linear classifier between each class and show that the variance is lower when using the proposed approach. The authors also compare against using a fixed augmentation strategy such as RandAugment and AUtoAugment and show that their method works better than using a fixed augmentation.

**Summary Of The Review:**

I really like the extensive experimentation and the different ablations done in the paper to show the efficacy of the approach. I am thus leaning towards accepting the paper.

---

> ### Author Response · Authors · 2022-11-11
> **Response to Reviewer URnx (2/2)**
>
> ---
> **Q3) When computing $V_{\text{correct}}$, is there an advantage of current uniformly random sampling compared to the case where the samples with larger (or smaller) losses are selected?**
>
> A3) Yes, there is an advantage of random sampling. This is because using randomly selected samples to compute $V_{\text{correct}}$ helps to extract the unbiased information for each class. If the small loss (or large loss) samples are selected, the augmentation strength increases too quickly (or slowly), resulting in performance degradation. Hence, we expected that the use of samples with small/large losses has lower performance than our random sampling. Based on this thought, we reported the results of a simple experiment on this as follows:
>
>
> |                        | category |  CE  | CE-DRW | LDAM-DRW |  BS  | RIDE |
> |------------------------|:--------:|:----:|:------:|:--------:|:----:|:----:|
> | Larger Loss            |   Many   | 67.0 |   61.6 |     63.1 | 59.9 | 67.9 |
> |                        |    Med   | 37.1 |   45.2 |     45.7 | 42.4 | 51.2 |
> |                        |    Few   |  7.3 |   20.3 |     20.3 | 23.3 | 25.8 |
> |                        |    ALL   | 38.6 |   43.5 |     44.2 | 42.8 | 49.4 |
> | Smaller Loss           |   Many   | 53.0 |   53.4 |     54.5 | 51.2 | 59.3 |
> |                        |    Med   | 24.7 |   33.0 |     33.9 | 36.1 | 38.4 |
> |                        |    Few   | 24.2 |   32.9 |     33.7 | 35.4 | 38.2 |
> |                        |    ALL   | 41.6 |   44.0 |     45.5 | 45.7 | 49.8 |
> | Random sampling (ours) |   Many   | 71.6 |   64.3 |     67.3 | 63.3 | 69.2 |
> |                        |    Med   | 42.3 |   49.2 |     50.4 | 48.4 | 52.8 |
> |                        |    Few   |  9.4 |   26.7 |     21.4 | 28.7 | 27.3 |
> |                        |    ALL   | 42.7 |   47.7 |     47.6 | 47.7 | 50.7 |
>
> As shown in the above table, when smaller or larger loss samples are used to measure $V_{\text{correct}}$, they suffer performance degradation. Therefore, it is a better way to grasp the degree of learning for each class without prejudice through uniform random sampling. Furthermore, computing loss for all samples for sorting them at the beginning of each epoch requires ***x1.5*** times more computation overhead than the proposed uniform random case. Thus, we strongly believe that random sampling is a better choice.

---

> ### Author Response · Authors · 2022-11-11
> **Response to Reviewer URnx (1/2)**
>
> We are very grateful for your constructive comments. We express your concerns that we understood and answer them as follows. If there is any worry that we misunderstood or additional questions, we look forward to being able to answer them with further comments from you.
>
> ---
> **Q1) Concern about the variance of performances from the randomness in subsampling for computing LoL**
>
> A1) We updated the standard deviation in the manuscript, and confirmed that the case of using CUDA (CE: 0.42, CE-DRW: 0.44, LDAM-DRW: 0.71, BS: 0.29, RIDE: 0.16, BCL: 0.20) is slightly larger than those of in the case of without CUDA (CE: 0.35, CE-DRW: 0.22, LDAM-DRW: 0.18, BS: 0.35, RIDE: 0.18). However, the standard deviation values for both cases are quite low, so we believe that the variance caused by subsampling is not significant. We additionally report the standard deviations for all results of our re-implementation of CIFAR-100-LT. We do not report the standard deviation values taken from another paper. All additional reported standard deviations verify statistical significance for the superiority of CUDA.
>
> ---
> **Q2) Suggestions on further analysis of “Dynamics of LoL score”**
>
> A2) To observe the dynamics of LoL score in the micro perspective, we check the dynamics of the top/bottom 10 classes for every 20 epochs by following the suggestion from the reviewer. We explain the observation after the table below.
>
>
> |          | Epoch     | 20  | 40  | 60  | 80  | 100 | 120 | 140 | 160 | 180 | 200 |
> |----------|-----------|-----|-----|-----|-----|-----|-----|-----|-----|-----|-----|
> | CE       | Top 10    | 0.5 | 0.8 | 1.7 | 2.0 | 2.6 | 2.7 | 3.2 | 3.2 | 3.4 | 3.7 |
> |          | Bottom 10 | 0.0 | 0.0 | 0.0 | 0.0 | 0.0 | 0.1 | 0.0 | 0.1 | 0.0 | 0.1 |
> | CE-DRW   | Top 10    | 0.8 | 1.3 | 2.0 | 2.5 | 1.6 | 2.6 | 1.8 | 2.6 | 2.5 | 2.6 |
> |          | Bottom 10 | 0.0 | 0.0 | 0.0 | 0.0 | 0.0 | 0.1 | 0.0 | 0.0 | 1.2 | 1.4 |
> | LDAM-DRW | Top 10    | 1.8 | 3.4 | 3.3 | 3.0 | 3.4 | 3.2 | 3.1 | 3.6 | 4.1 | 4.3 |
> |          | Bottom 10 | 0.0 | 0.0 | 0.1 | 0.0 | 0.0 | 0.1 | 0.0 | 0.0 | 1.8 | 1.4 |
> | BS       | Top 10    | 1.1 | 1.1 | 1.6 | 2.4 | 2.1 | 2.5 | 2.5 | 2.8 | 3.9 | 4.0 |
> |          | Bottom 10 | 0.1 | 0.1 | 0.0 | 0.1 | 0.6 | 1.1 | 1.0 | 0.7 | 1.7 | 1.4 |
> | RIDE     | Top 10    | 0.8 | 1.2 | 1.6 | 1.7 | 2.1 | 1.8 | 1.3 | 1.1 | 2.5 | 2.6 |
> |          | Bottom 10 | 0.0 | 0.0 | 0.0 | 0.0 | 0.0 | 0.0 | 0.0 | 0.0 | 0.7 | 0.8 |
>
> We summarize the analysis of the dynamics over time, which is noted in the manuscript (page 9) and further analysis at the end time as follows:
> - **(Dynamics over time)**
> CE: When we naively train the model, CUDA allocates high and low strength augmentation to “many” and “few” categories, respectively, which make us expect to achieve high balanced performance based on our main findings (Figure 1).
> Balanced Loss (BS): In the case of BS, CUDA allocates the similar augmentation strength to both categories, since BS has an adjustment module to reduce and enlarge the impact of “many” and “few” categories, respectively.
> Two-stage training (*-DRW): As in the last phase (>160 epochs) of training (i.e., when balancing weights are applied, LoL scores of the “few” category start to increase). This is because the classes in the “few” category are learned by the model thanks to the reweighting.
> - **(End time)**
> CE: Because CE does not have a rebalancing mechanism, it fails to learn the “few” classes by seeing their LoL score.
> Others: When the model is trained with LTR methods, the LoL score gap between the “few” and “many” categories are smaller which shows that rebalancing methods encourage minority classes.
> For all methods, as the final LoL score of the “many” category is larger than the “few” category, we can conclude that CUDA improves the minority classes by providing stronger augmentation to majority classes, which is consistent with our finding in Figure 1.

---

### Official Review · Reviewer_N4yQ · 2022-10-25

**Confidence:** 4
**Correctness:** 3
**Technical Novelty And Significance:** 3
**Empirical Novelty And Significance:** 3
**Recommendation:** 6

**Clarity, Quality, Novelty And Reproducibility:**

The motivation of this paper is novel and interesting.

In this paper, 22 augmentations are considered. I am curious if the phenomenon that "Our key finding is that class-wise augmentation improves performance in the non-augmented classes while that for the augmented classes may not be significantly improved, and in some cases, performances may even decrease." only exists in this setting. If we only consider several simple augmentations, for example, flip and resize, will the conclusion still satisfy?

**Strength And Weaknesses:**

Strength:

1. "Our key finding is that class-wise augmentation improves performance in the non-augmented classes while that for the augmented classes may not be significantly improved, and in some cases, performances may even decrease." The motivation and the findings in this paper are really interesting.

2. The results are good and comprehensive in the experiments.

3. The paper is well-presented and easy to follow.

Weakness:

1. One of the key ablations is missing. What are the results of uniformly assigning all data augmentations to all classes? This should be a baseline and presented in the tables. That is, the conventional operation

2. Will the order of different augmentations have an effect on the final results?

3. What is the effect of preset augmentation number K?


**Summary Of The Paper:**

This paper found that augmentations in one class may be negative for itself but positive for other classes. Thus, they propose a novel data augmentation method, call CUDA for long-tailed recognition. This method can generate proper class-wise augmentation strength for long-tailed recognition. They first compute a level-of-learning score for each class and leverage the score to determine the augmentation. The authors conduct experiments on cifar-lt, imagenet-lt and inaturalist 2018. The comparisons with previous SOTA methods demonstrate the effectiveness of CUDA.

**Summary Of The Review:**

The motivation, writing, and the results are good of this paper. However, there are some points which need further clarification.

---

> ### Author Response · Authors · 2022-11-11
> **Response to Reviewer N4yQ (3/3)**
>
>
> ---
>
> **Q4) When the number of preset augmentation decreases, does the finding in Figure 1 “the performance of specific class decreases when strong augmentation is applied to that class” consistently occur?**
>
> A4) Yes, the finding in Figure 1 will happen if there are enough augmentation operations to make the samples different enough. To verify this, we report the experimental results with the configurations with 10 operations which are introduced in A3. The performance of (0,0), (0,4), (4,0), and (4,4) that each configuration denotes the augmentation strength of (majority; top 50 classes, minor; bottom 50 classes) are described as follows:
>
>
> |          | (major, minor) | Many |  Med |  Few |  All |
> |----------|:--------------:|:----:|:----:|:----:|:----:|
> | CE       |       0,0      | 66.2 | 37.3 |  8.2 | 38.7 |
> |          |       0,4      | **69.7** | 30.4 |  2.3 | 35.7 |
> |          |       4,0      | 60.9 | 39.3 | **12.8** | 38.9 |
> |          |       4,4      | 67.0 | 34.5 |  4.7 | 37.0 |
> | CE-DRW   |       0,0      | 62.8 | 41.7 | 16.2 | 41.4 |
> |          |       0,4      | **65.9** | 37.2 | 10.6 | 39.3 |
> |          |       4,0      | 49.3 | 45.2 | **28.3** | 41.6 |
> |          |       4,4      | 56.6 | 46.6 | 24.6 | 43.5 |
> | LDAM-DRW |       0,0      | 62.8 | 42.3 | 19.0 | 42.5 |
> |          |       0,4      | **70.1** | 34.3 |  6.4 | 38.5 |
> |          |       4,0      | 52.3 | 42.0 | **27.7** | 41.3 |
> |          |       4,4      | 61.1 | 43.4 | 17.3 | 41.8 |
> | BS       |       0,0      | 61.6 | 42.3 | 23.0 | 43.3 |
> |          |       0,4      | **66.9** | 37.9 | 10.8 | 39.9 |
> |          |       4,0      | 48.7 | 42.5 | **28.7** | 40.5 |
> |          |       4,4      | 56.3 | 44.2 | 23.0 | 42.1 |
> | RIDE     |       0,0      | 67.7 | 51.5 | 26.7 | 49.7 |
> |          |       0,4      | **70.5** | 36.8 |  7.7 | 39.9 |
> |          |       4,0      | 56.6 | 44.5 | **27.2** | 43.6 |
> |          |       4,4      | 62.3 | 44.5 | 21.6 | 43.9 |
>
> As shown in the table above, when majorities are transformed with strong augmentation, i.e., (4,0), we can see that minorities have higher performance than the (0,0) case. On the contrary, when the augmentation of the minorities is strengthened (0,4), the performance of the majorities is improved. Through this, we argue that the finding in Figure 1 is valid even in a small number (e.g., $K=10$) of predefined augmentation operations.

---

> > ### Comment · Reviewer_N4yQ · 2022-11-25
> > **Response to rebuttal**
> >
> > I think the authors have well-addressed my concerns in the rebuttal.
> >
> > The motivation for different augmentation for different classes is interesting.
> >
> > During the rebuttal discussion, the paper is further improved. Thus, I raise my score to 6.
> >
> > A little suggestion for the colour of highlights in the revision: the adopted red seems not so obvious, the blue may be better?

---

> > > ### Author Response · Authors · 2022-11-25
> > > **Thank you for your response!**
> > >
> > > Dear Reviewer N4yQ,
> > >
> > > We really thank you for spending your time to give us constructive comments.
> > >
> > > We are happy to address your concerns and to make our contribution clearer.
> > >
> > > Regarding your comment on the highlighting, we are sorry that it is impossible for us to modify the manuscript in the second phase.
> > >
> > > Instead, we will consider your comment in future revisions.
> > >
> > > Sincerely,
> > >
> > > Authors

---

> ### Author Response · Authors · 2022-11-11
> **Response to Reviewer N4yQ (2/3)**
>
>
> ---
> **Q3) What is the effect of the size of the augmentation preset?**
>
> A3) CUDA works well if the size of the augmentation preset $K$ is enough to generate difficult samples. In our experiments, we confirmed that the 10 randomly selected augmentation presets also show sufficient performance gain. The following are our experimental settings.
> We evaluate the two cases of augmentation preset: (1) 10 randomly selected augmentations {Mirror, ShearX, Invert, Smooth, ResizeCrop, Color, Brightness, Sharpness, Rotate, AutoContrast} and (2) RandAugment preset {AutoContrast, Equalize, Invert, Rotate, Posterize, Solarize, SolarizeAdd, Color, Contrast, Brightness, Sharpness, ShearX, ShearY, CutoutAbs, TranslateXabs, TranslateYabs}.
>
>
> |                                | category |  CE  | CE-DRW | LDAM-DRW |  BS  | RIDE |
> |--------------------------------|:--------:|:----:|:------:|:--------:|:----:|:----:|
> | Vanilla (without augmentation) | Many     | 66.2 |   62.8 |     62.8 | 61.6 | 67.7 |
> |                                | Med      | 37.3 |   41.7 |     42.3 | 42.3 | 51.5 |
> |                                | Few      |  8.2 |   16.2 |     19.0 | 23.0 | 26.7 |
> |                                | All      | 38.7 |   41.4 |     42.5 | 43.3 | 49.7 |
> | Randomly selected preset ($K=10$)             | Many     | 70.8 |   62.3 |     65.2 | 62.6 | 68.5 |
> |                                | Med      | 40.4 |   49.0 |     49.2 | 46.9 | 52.0 |
> |                                | Few      |  9.0 |   26.7 |     21.6 | 27.3 | 27.1 |
> |                                | All      | 41.6 |   47.0 |     46.5 | 46.5 | 50.3 |
> | RandAugment preset($K=14$)               | Many     | 70.3 |   63.5 |     65.4 | 62.9 | 68.5 |
> |                                | Med      | 40.7 |   49.1 |     50.6 | 48.1 | 52.3 |
> |                                | Few      |  9.6 |   26.0 |     21.6 | 28.7 | 27.0 |
> |                                | All      | 41.8 |   47.2 |     47.1 | 47.5 | 50.4 |
> | CUDA  ($K=22$)              | Many     | 71.6 |   64.3 |     67.3 | 63.3 | 69.2 |
> |                                | Med      | 42.3 |   49.2 |     50.4 | 48.4 | 52.8 |
> |                                | Few      |  9.4 |   26.7 |     21.4 | 28.7 | 27.3 |
> |                                | All      | 42.7 |   47.7 |     47.6 | 47.7 | 50.7 |
>
> The aboveable shows that the accuracy slightly increases when the size of the augmentation preset increases. However, the gap between the RandAugment preset (14 operations) and our original preset (22 operations) is small compared to that between the vanilla (without CUDA case) and the RandAugment case. These results verify our belief that the impact of the number of predefined augmentations is small.

---

> ### Author Response · Authors · 2022-11-11
> **Response to Reviewer N4yQ (1/3)**
>
> We are very grateful for your constructive comments. We express your concerns as we understood and answer them as follows. If there is any worry that we misunderstood or additional questions, we look forward to being able to answer them with further comments from you.
>
> ---
>
> **Q1) What if CUDA is applied equally to all classes other than class-wisely?**
>
> A1) If CUDA is not class-wise (i.e., if all classes are given the same augmentation strength) performance degradation occurs. This is because the proper augmentation, generating the most difficult sample without losing its original information, for each class is very different among classes.
>
> To evaluate the case of all classes being augmented with the same augmentation strength based on CUDA, we conduct the experiments on the CIFAR-100-LT with an imbalance ratio of 100 and describe the results in the following table:
>
>
> |                     | category | CE          | CE-DRW      | LDAM-DRW    | BS          | RIDE        |
> |---------------------|----------|-------------|-------------|-------------|-------------|-------------|
> | CUDA w/o class-wise | Many     |        69.0 |        62.7 |        65.2 |        62.2 |        67.7 |
> |                     | Med      |        38.7 |        47.3 |        47.6 |        44.7 |        52.2 |
> |                     | Few      |         6.7 |        23.5 |        20.7 |        25.5 |        26.4 |
> |                     | All      |        39.7 |        45.5 |        45.7 |        44.9 |        49.9 |
> | Vanilla             | Many     | 66.2 (-2.8) | 62.8 (+0.1) | 62.8 (-2.4) | 61.6 (-0.6) | 67.7 (+0.0) |
> |                     | Med      | 37.3 (-1.4) | 41.7 (-5.6) | 42.3 (-5.3) | 42.3 (-2.4) | 51.5 (-0.7) |
> |                     | Few      |  8.2 (+1.5) | 16.2 (-7.3) | 19.0 (-1.7) | 23.0 (-2.5) | 26.7 (+0.3) |
> |                     | All      | 38.7 (-1.0) | 41.4 (-4.1) | 42.5 (-3.2) | 43.3 (-1.6) | 49.7 (-0.2) |
> | CUDA (Ours)         | Many     | 71.6 (+2.6) | 64.3 (+1.6) | 67.3 (+2.1) | 63.3 (+1.1) | 69.2 (+1.5) |
> |                     | Med      | 42.3 (+3.6) | 49.2 (+1.9) | 50.4 (+2.8) | 48.4 (+3.7) | 52.8 (+0.6) |
> |                     | Few      |  9.4 (+2.7) | 26.7 (+3.2) | 21.4 (+0.7) | 28.7 (+3.2) | 27.3 (+0.9) |
> |                     | All      | 42.7 (+3.0) | 47.7 (+2.2) | 47.6 (+1.9) | 47.7 (+2.8) | 50.7 (+0.8) |
>
> Note that values in parentheses are the gain from the performance of CUDA w/o class-wise (i.e., vanilla - CUDA w/o class-wise, CUDA - CUDA w/o class-wise). As shown in the above table, the performances of the few categories are significantly degraded; in particular, the case of All performance is also degraded compared to the class-wise approach.
>
> ---
> **Q2) Will the different augmentation orders have an impact on the result?**
>
> A2) The augmentation sequence has a minor impact on CUDA performance. This is because CUDA focuses on the difficulty level of samples, which is sufficiently controlled by the number of augmentation operations and their magnitudes. To confirm this, we report the performance of the case where the order of augmentation operations is fixed. For example, when the operation indices (6, 3, 5) among 22 augmentations are sampled, it is applied with sorted order, i.e., (3, 5, 6).
>
> |             | Category |      CE     |    CE-DRW   |   LDAM-DRW  |      BS     |     RIDE    |
> |-------------|:--------:|:-----------:|:-----------:|:-----------:|:-----------:|:-----------:|
> | CUDA (Ours) | Many     |        71.6 |        64.3 |        67.3 |        63.3 |        69.2 |
> |             | Med      |        42.3 |        49.2 |        50.4 |        48.4 |        52.8 |
> |             | Few      |         9.4 |        26.7 |        21.4 |        28.7 |        27.3 |
> |             | All      |        42.7 |        47.7 |        47.6 |        47.7 |        50.7 |
> | Fixed order | Many     | 70.5 (-1.1) | 62.8 (-1.5) | 66.9 (-0.4) | 62.5 (-0.8) | 68.2 (-1.0) |
> |             | Med      | 43.0 (+0.7) | 50.2 (+1.0) | 49.7 (-0.7) | 48.3 (-0.1) | 53.5 (+0.7) |
> |             | Few      |  9.0 (-0.4) | 27.0 (+0.3) | 21.9 (+0.5) | 29.6 (+0.9) | 26.9 (-0.4) |
> |             | All      | 42.4 (-0.3) | 47.7 (+0.0) | 47.4 (-0.2) | 47.7 (+0.0) | 50.5 (-0.2) |
>
> Note that the values in parentheses represent (Fixed order - CUDA). As shown in the above table, we argue that the effect of the augmentation order on the difficulty is negligible, and that thus the effect on the performance of CUDA is minimal.

---

> ### Author Response · Authors · 2022-11-23
> **Additional experiments and respectful re-evaluation request**
>
> We expect that our previous answers addressed the concerns of the reviewer N4yQ. We report additional experiments below (when the size of the augmentation preset is extremely small, such as K=2) with the hope that these results help the reviewer to re-evaluate this paper.
>
> This experiment is conducted with only two predefined augmentation operations (flip and resize, as suggested by the reviewer N4yQ) to make sure that the effect of per-class augmentation. Same as A4, the performances of (0, 0), (2, 0), (0, 2), and (2, 2) that each configuration denotes the augmentation strength of (M: top 50 classes, m: bottom 50 classes) are described as follows:
>
> | Method | M/m | Many | Med  | Few  | All  |
> |----|-----|------|------|------|------|
> | CE   | 0,0 | 66.2 | 37.3 | 8.2  | 38.7 |
> |    | 2,0 | 64.0 | 38.5 | **9.7**  | 38.8 |
> |    | 0,2 | **66.9** | 34.4 | 5.83 | 37.2 |
> |    | 2,2 | 65.1 | 36.2 | 7.4  | 37.7 |
> |CE-DRW  | 0,0 | 62.8 | 41.7 | 16.2 | 41.4 |
> |        | 2,0 | 57.7 | 42.9 | **19.4** | 41.1 |
> |        | 0,2 | **64.2** | 28.4 | 7.8  | 34.8 |
> |        | 2,2 | 45.3 | 30.7 | 16.2 | 31.5 |
> |LDAM-DRW| 0,0 | 62.8 | 42.3 | 19.0 | 42.5 |
> |          | 2,0 | 60.0 | 42.7 | **22.5** | 42.7 |
> |          | 0,2 | **64.6** | 38.3 | 10.5 | 39.2 |
> |          | 2,2 | 61.6 | 41.7 | 18.1 | 41.6 |
> |BS | 0,0 | 61.6 | 42.3 | 23.0 | 43.3 |
> |    | 2,0 | 55.2 | 42.5 | **26.9** | 42.3 |
> |    | 0,2 | **62.5** | 29.0 | 10.9 | 35.3 |
> |    | 2,2 | 54.0 | 41.6 | 25.2 | 41.0 |
> |RIDX| 0,0 | 67.7 | 51.5 | 26.7 | 49.7 |
> |      | 2,0 | 61.0 | 47.1 | **27.9** | 46.2 |
> |      | 0,2 | **70.9** | 43.1 | 16.3 | 44.8 |
> |      | 2,2 | 61.7 | 45.7 | 22.3 | 44.3 |
>
> As shown in the table, we can see that our finding in Figure 1 also holds with an **extremely** small number ($K=2$) of predefined augmentation operations.

---

### Official Review · Reviewer_Z6g5 · 2022-10-25

**Confidence:** 4
**Clarity, Quality, Novelty And Reproducibility:** Please see the comments in the previo…
**Correctness:** 3
**Technical Novelty And Significance:** 3
**Empirical Novelty And Significance:** 3
**Recommendation:** 6

**Strength And Weaknesses:**

Pros:
1. The main observation of the paper- "class-wise augmentation actually improves the performance of non-augmented classes" is very crucial and important.
2. The data augmentation strategy devised by the authors also seems clever and logical. However, there are a few things that are concerning (see concerns).

Concerns:
1. The way DA with strength parameter is devised is concerning because it assumes that the sequence of augmentation is permutation invariant and preserves the identity of the original sample. This need not be the case with many augmentations.
2. From Fig. 5c it seems that the method is sensitive to threshold / accept rate \gamma, in experiments also, the authors choose 2 different values of \gamma (0.4 and 0.6). What procedure do the authors follow to decide this value? Is there any strategy one can follow to come up with this value?
3. In Table 2, the improvement with CUDA looks very incremental.
4. For such close values it is suggested that the authors report the scores in the form of mean \pm std. dev to give a more clear view of how close the scores actually are.
5. The authors show that the variance of L1-norm of a linear classifier decreases due to CUDA. In general, any data-augmentation strategy should decrease the variance because of an increase in the data samples.



**Summary Of The Paper:**

The paper proposes an auxiliary data augmentation technique that can be used on top of several long-tailed recognition methods. The main insight of the paper is that class-wise augmentation actually improves the performance of non-augmented classes. This is an important as well as counter-intuitive observation. The authors devise a data augmentation strategy that relies on the `strength' of augmentation, which is calculated using level-of-learning.


**Summary Of The Review:**

Overall, I feel the paper has a good amount of novelty and the observations made are crucial for the community. However, there are some concerns that need to be answered before going forward. Hence, at the moment I will keep my score as borderline reject. I would like to see other reviewers' scores and author rebuttals.

---

> ### Author Response · Authors · 2022-11-11
> **Response to Reviewer Z6g5 (3/3)**
>
>
> ---
>
> **Q4) Concerns about the claim that “reducing the variance of the L1 norm of a linear classifier is not a gain of CUDA, because it is usually reduced by data augmentation”**
>
> A4) We would like to emphasize that reducing the variance of the L1 norm in a linear classifier is more pronounced in CUDA than in general augmentation. More specifically, CUDA mitigates the variance in the class imbalance problem by adjusting the strength of data augmentation. To confirm this claim, we describe two analyses: (1) by using what we described in Appendix A. 2 (i.e., Figure 7) and (2) the result of CE-DRW under three settings: without augmentation, CUDA with AutoAugmentation searched on CIFAR dataset, and CUDA.
>
> (1) When we utilize data augmentation for a class imbalance dataset, the L1 norm distribution of the trained model is depicted in Appendix A. 2. We summarize the figure by using the following table. Here, Normal, Augment all, and  Augment (0-49) represent standard augmentation (random crop and horizontal flip), strong augmentation (strength 4) to all classes equally, and strong augmentation (4) to the top 50 classes.
>
> |                                           | Normal (Standard augment) | Augment All | Augment (0-49)  |
> |-------------------------------------------|:-----------:|:-----------:|:---------------:|
> | Weight L1 norm variance (end of training) |    2.630    |    1.264    |      0.809      |
>
>
> We measure the variance of the L1 norm of the trained linear classifier (i.e., after 200 epochs training). As shown in the above table, despite giving augmentation to all classes equally can reduce variance, partial augment reduces further, thus it can make the linear classifier be fair.
>
> (2) We utilize the same setting in the manuscript and CIFAR-100-LT with an imbalance ratio 100 dataset. We describe the variance of L1 norm weight every 20 epochs, as follows:
>
> | Epoch   | 0     | 20    | 40    | 60    | 80    | 100   | 120   | 140   | 160   | 180   | 200   |
> |---------|-------|-------|-------|-------|-------|-------|-------|-------|-------|-------|-------|
> | w/o Aug | 0.826 | 3.018 | 3.448 | 3.470 | 3.342 | 3.202 | 3.093 | 3.104 | 3.044 | 2.967 | 2.967 |
> | AutoAug | 0.819 | 2.700 | 3.149 | 3.065 | 2.958 | 2.908 | 2.810 | 2.723 | 2.646 | 2.527 | 2.526 |
> | CUDA    | 0.821 | 2.651 | 2.892 | 2.708 | 2.502 | 2.444 | 2.393 | 2.364 | 2.297 | 2.182 | 2.182 |
>
>
> As shown in the table above, CUDA reduces the variance of the L1 norm of a linear classifier compared to the AutoAugment augmentation policy. This means that strength control for each class based on how well trained they are is an important part of CUDA for making a linear classifier fair.
>
> ---
> [A] AutoAugment: Learning Augmentation Policies from Data. CVPR. 2019
>
> [B] RandAugment: Practical automated data augmentation with a reduced search space. NeurIPS. 2020
>
> [C] Long-tailed Recognition by Routing Diverse Distribution-Aware Experts. ICLR. 2021
>
> [D] The Majority Can Help the Minority: Context-rich Minority Oversampling for Long-tailed Classification. CVPR. 2022
>
> [E] Balanced Contrastive Learning for Long-Tailed Visual Recognition. CVPR. 2022

---

> > ### Comment · Reviewer_Z6g5 · 2022-11-18
> > **Response on the Revision.**
> >
> > I read the Authors' response and the revised manuscript. It looks much better now with many of the reviewers' concern addressed. I increase my score to an accept now.

---

> > > ### Author Response · Authors · 2022-11-18
> > > **Thank you for your positive assessment**
> > >
> > > Dear Reviewer Z6g5,
> > >
> > > We deeply appreciate you taking the time to read our response.
> > >
> > > Thanks to you, we were able to create a manuscript that could be delivered more clearly.
> > >
> > >
> > > Sincerely,
> > >
> > > Authors

---

> ### Author Response · Authors · 2022-11-11
> **Response to Reviewer Z6g5 (2/3)**
>
>
> ---
> **Q3) Concerns about low performance gain in a large-scale benchmark (Table 2)**
>
> A3) We think that our improvement is comparable, by seeing the performance of the previous papers, such as RIDE [C], CMO [D], and BCL [E]. We summarize the performance gain of each algorithm in our implementation as follows:
>
> |             | CMO (CVPR’22)                          | BCL (CVPR’22)                     | CUDA (Ours)                            |
> |-------------|----------------------------------------|-----------------------------------|----------------------------------------|
> | ImageNet-LT | (RIDE -> CMO+RIDE)53.6 -> 54.0 (+0.4)  | (RIDE -> BCL)53.6 -> 55.6 (+2.0)  | (BCL-> CUDA+BCL) 55.6 -> 56.3 (+0.7)   |
> | iNaturalist | (RIDE -> CMO+RIDE) 71.6 -> 70.9 (-0.7) | (RIDE -> BCL) 71.6 -> 71.8 (+0.2) | (BCL -> CUDA+RIDE) 71.8 -> 72.4 (+0.6) |
>
>
> As shown in the table above, the gain of CUDA on ImageNet-LT (from 55.6 to 56.3 – i.e., +0.7) and iNaturalist (from 71.8 to 72.4 – i.e., +0.6) is comparable to the gain of previous works. For example, CMO and BCL obtain 0.4% and 1.4%, respectively, on ImageNet-LT. Furthermore, in the iNaturalist case, CMO shows -0.7% degradation, and BCL obtains 0.6% improvement compared to RIDE on our implementation. Note that the official code of CMO did not contain CMO+RIDE case; thus, we re-implement it by utilizing the case of CMO+* (e.g., CMO+BS). Therefore, we believe that the gain of CUDA is meaningful.
>
> In addition, to answer the concern about the sensitivity of large-scale datasets, we added standard deviation values to Table 1 and Table 2 in the revised manuscript. As in the ImageNet-LT part in Table 2, the standard deviation of CUDA is at most 0.2. For the iNaturalist 2018 dataset, the largest in our experiments, we couldn’t report all standard deviations due to the limitation of time. Instead, we report the standard deviation values of RIDE ($71.6 \pm 0.1$) and RIDE+CUDA ($72.4 \pm 0.2$), which show statistical significance for the superiority of our method. We are planning to report all standard deviation values for iNaturalist 2018 before the end of the first-phase rebuttal. From the additionally reported standard deviations, we believe that, when we apply CUDA to the baselines on the large-scale benchmarks, their sensitivity is not significant and the gains of CUDA are valid.
>
> ---
>
> **Additional note (Standard deviation) 2022.11.15 (05:30 AM UTC)**
>
> We have updated the remaining standard deviation results on the iNaturalist 2018 dataset.

---

> ### Author Response · Authors · 2022-11-11
> **Response to Reviewer Z6g5 (1/3)**
>
> We are very grateful for your constructive comments. We have re-written your comments as we understand them and provide corresponding answers. Please let us know if there are any misunderstandings or if you have additional questions.
>
> ---
> **Q1) When CUDA transforms the given image with an augmentation strength parameter, does it assume that the augmentation operation should contain information of the original image and that the operation is permutation invariant?**
>
> A1) In short, we did not assume permutation invariance, and we designed CUDA so that the augmentation operation can contain information on the original image like other augmentation papers, such as AutoAugment [A] and RandAugment [B].
> Permutation invariance: CUDA does not require any assumptions about the order of augmentation operations. For example, Rotate $\to$ Brighten and Brighten $\to$ Rotate can be different. CUDA randomly decides the augmentation order and does not consider it to adjust the difficulty.
> Preserving original information: The original information on the image is necessary for the training procedure. We thus design CUDA so that it can maintain information from the input image through a pre-defined magnitude function, similar to other studies such as AutoAugment and RandAugment. For example, the maximum rotation is limited to 30 degrees.
>
> ---
>
> **Q2) The sensitivity of the hyperparameter $\gamma$ and the tuning policy that determines $\gamma$.**
>
> A2) For this question, we answer as follows:
> - **(Sensitivity)** We would like to highlight that CUDA consistently improves validation performance regardless of hyper-parameter selection. In Figure 5, CUDA with all $\gamma$ values in the range of (0.0, 1.0) is better than the LTR method (RIDE).
> Moreover, we believe $\gamma$ is not sensitive. The performance of RIDE has values from 50.55 to 50.75 for the $\gamma$ range $[0.5,0.7]$, reported in Figure 5 (c).
>
> - **(Tuning strategy)** We decide the accept rate $\gamma$ from the training behavior with a simple rule. Our strategy for tuning $\gamma$ is to select the largest value in which at least one of LoL scores among all classes increases within 20 epochs. There could be a much better strategy to find the best $\gamma$, but this is beyond the scope of this work.
> Here is the detailed tuning strategy of $\gamma$:
> We initially set $\gamma$ as 0.6. However, for large-scale datasets, they fail to infer even the easier-to-learn majority classes because they are difficult to learn due to the huge size of the training dataset. Therefore, we decrease the threshold $\gamma$ by 0.1 points whenever it fails to raise any of LoL scores for the first 20 training epochs. We conduct this search on CE with CIFAR-100-LT with imbalance ratio (IR) 100 and using the same $\gamma$ value for the other algorithms with remaining IR settings. Also, we conduct this search rule on ImageNet-LT with CE and use the same value with the other large-scale dataset (iNaturalist 2018) with remaining algorithms.
>
>
>
> We additionally conduct the sensitivity analysis of $\gamma \in$ {0.3, 0.4, 0.5, 0.6} with Balanced Softmax (BS) on ImageNet-LT. The results are follows:
>
> |          |   0.3 |   0.4 |   0.5 |   0.6 |
> |----------|------:|------:|------:|------:|
> | Acc. (%) | 51.42 | 51.59 | 51.38 | 51.24 |
>
> While our selected value (i.e., $\gamma = 0.4$) achieved the best performance, all performances are higher than the baseline, BS (50.9%).

---

### Author Response · Authors · 2022-11-11
**General Response**

Dear reviewers,

We wish to thank all reviewers (**@Z6g5, @N4yQ, @URnx, @QwML**) sincerely for carefully reviewing our paper and giving insightful and constructive comments. We are genuinely encouraged to find interesting observations (**@Z6g5, @N4yQ, @URnx, @QwML**) that class-wise augmentation actually improves the performance of non-augmented classes and novel curriculum of data augmentation method (**@Z6g5, @URnx, @QwML**) with extensive experiments (**@N4yQ, @URnx, @QwML**) for class imbalance problem. Furthermore, we are pleased that many reviewers (**@N4yQ, @URnx, @nPVy, @QwML**) feel our paper is well-organized, clearly written, and easy to understand.

We addressed the concerns from reviewers in each reply, and modified our manuscript reflecting this as follows:

---
- Much clearer description of Figure 1 for @nPVy and @QwML (Section 1)
- Standard deviation in Table 1 and 2, for @Z6g5 and @URnx (Section 4)
- Hyperparameter tuning strategy, for @Z6g5 and @nPVy (Appendix B.4)
- Further analysis on augmentation preset, for @N4yQ and @QwML (Appendix D.2)
- Detailed experimental settings of Figure 5(d), for @QwML (Appendix E)
- Further analysis on LoL Score, for @URnx and @QwML (Appendix F.1)
- Analysis the case of without class-wise, for @N4yQ and @nPVy (Appendix F.2)
---

The revisions made are marked with $\color{purple}{\text{purple}}$ colors in the revised paper.

Thank you for your time.

Sincerely,


Authors

---

### Author Response · Authors · 2022-11-18
**The end of the first discussion phase is approaching**

We appreciate the reviewers' constructive reviewing of our paper and insightful comments.


We revised our manuscript based on your first comments and we hope that your concerns have been addressed.


After the end of the first discussion phase (**Nov. 18**), the manuscript cannot be revised, as you already know.


Therefore, during the second phase, we will address your additional concerns and incorporate them into the final draft.


If you have any further inquiries, please feel free to contact us.


*We look forward further constructive discussion.*

Thank you.

Sincerely,

Authors

---

### Public Comment · ~Zhihan_Zhou2 · 2023-02-13
**Congratulations to the nice work! Recommend our work on self-supervised long-tailed learning.**

Hi,

Congratulations to the nice work! Our recent ICML'22 paper BCL (https://proceedings.mlr.press/v162/zhou22l/zhou22l.pdf) also studied long-tailed learning from the perspective of data augmentation, but in a self-supervised manner. Would you mind adding a discussion about our work? Many thanks!

Best,

Zhihan

---

> ### Author Response · Authors · 2023-02-15
> **Response**
>
> Dear Zhihan Zhou,
>
> Thank you for your congratulations.
>
> We will check out your recommended papers and properly cite them in our camera-ready version.
>
> Best,
> Sumyeong Ahn.

---

### Decision · Program_Chairs · 2023-01-20

**Decision:**

Accept: notable-top-25%

**Justification For Why Not Higher Score:**

This paper executes well on an interesting observation. To reach the level of oral presentation, more should be said to make it appealing for the general audience: either provide more rigorous principles, very strong results, or wider applications to various domains and problems.

**Justification For Why Not Lower Score:**

(1) An interesting observations and idea. (2) good experimental results.
I recommend spotlight because it may open future research in other areas where data augmentation is very useful (few shot learning, contrastive learning, etc).


**Metareview: Summary, Strengths And Weaknesses:**

The paper describes a data augmentation technique for long-tailed recognition, which is based on the following observation: Augmenting data from some classes may hurt that class and improve the performance of other, non-augmented classes.
Then, based on that observation, they compute a per-class score ("level of learning") used to determine a "curriculum" for the augmentation (which they name CUDA). Experiments show that adding CUDA, to existing augmentation techniques, provides good results on CIFAR-LT, ImageNet-LT, and iNaturalist.

After a detailed rebuttal, where authors improved the paper through clarifications and experiments, all four reviewers recommended acceptance.

It would be interesting to understand if similar curriculum techniques can be used in learning setups beyond long-tail, where data augmentation is also critical (few shot learning, contrastive learning, generative AI etc). Also, what can be said about sample efficiency of the proposed augmentation policy?

**Note From Pc:**

if the above contains the word "oral" or "spotlight" please see: "oral" presentation means -> notable-top-5% and "spotlight" means -> notable-top-25%. As stated in our emails, we are disassociating presentation type from AC recommendations

**Summary Of Ac-Reviewer Meeting:**

N/A